# Allosteric activation of the nitric oxide receptor soluble guanylate cyclase mapped by cryo-electron microscopy

Benjamin G Horst[1†], Adam L Yokom[2,3†], Daniel J Rosenberg[4,5], Kyle L Morris[2,3‡], Michal Hammel[4], James H Hurley[2,3,4,5*], Michael A Marletta[1,2,3*]

[1]Department of Chemistry, University of California, Berkeley, Berkeley, United States; [2]Department of Molecular and Cell Biology, University of California, Berkeley, Berkeley, United States; [3]Graduate Group in Biophysics, University of California, Berkeley, Berkeley, United States; [4]Molecular Biophysics and Integrated Bioimaging, Lawrence Berkeley National Laboratory, Berkeley, United States; [5]California Institute for Quantitative Biosciences, University of California, Berkeley, Berkeley, United States

**Abstract** Soluble guanylate cyclase (sGC) is the primary receptor for nitric oxide (NO) in mammalian nitric oxide signaling. We determined structures of full-length *Manduca sexta* sGC in both inactive and active states using cryo-electron microscopy. NO and the sGC-specific stimulator YC-1 induce a 71° rotation of the heme-binding β H-NOX and PAS domains. Repositioning of the β H-NOX domain leads to a straightening of the coiled-coil domains, which, in turn, use the motion to move the catalytic domains into an active conformation. YC-1 binds directly between the β H-NOX domain and the two CC domains. The structural elongation of the particle observed in cryo-EM was corroborated in solution using small angle X-ray scattering (SAXS). These structures delineate the endpoints of the allosteric transition responsible for the major cyclic GMP-dependent physiological effects of NO.

DOI: https://doi.org/10.7554/eLife.50634.001

**\*For correspondence:**
jimhurley@berkeley.edu (JHH);
marletta@berkeley.edu (MAM)

[†]These authors contributed equally to this work

**Present address:** [‡]MRC London Institute of Medical Sciences, London, United Kingdom

## Introduction

Nitric oxide (NO) is a critical primary signaling molecule in eukaryotic organisms (*Palmer et al., 1987*; *Moncada et al., 1991*). Cells detect this diatomic gas through the NO receptor soluble guanylate cyclase (sGC), which is activated by NO to catalyze the cyclization of 5'-guanosine triphosphate (GTP) to 3',5'-cyclic guanosine monophosphate (cGMP) (*Arnold et al., 1977*; *Stone and Marletta, 1994*; *Russwurm and Koesling, 2004*; *Derbyshire and Marletta, 2012*; *Montfort et al., 2017*; *Horst and Marletta, 2018*). The NO signaling pathway, with sGC as a central component, plays crucial roles in the cardiovascular and neurological systems in mammals (*Lucas et al., 2000*; *Prast and Philippu, 2001*; *Benarroch, 2011*). Furthermore, aberrations in these signaling pathways can lead to pathologies that include various forms of hypertension, cardiovascular disease, and neurodegeneration (*Bredt, 1999*; *Friebe and Koesling, 2009*; *Lundberg et al., 2015*; *Hervé et al., 2014*; *Wallace et al., 2016*).

sGC is a heterodimer composed of two subunits, denoted α and β, that are each composed of four domains: an N-terminal heme nitric oxide/oxygen (H-NOX) domain, a Per/Arnt/Sim (PAS)-like domain, a coiled-coil (CC) domain, and a catalytic (CAT) domain (*Derbyshire and Marletta, 2012*; *Montfort et al., 2017*). Although each subunit contains an H-NOX domain, only the β H-NOX domain binds a heme cofactor, with direct ligation occurring through a conserved histidine residue (*Zhao et al., 1998*). The CC domains, together with the PAS domains, are thought to form a

**eLife digest** In humans and other animals, as the heart pumps blood around the body, the blood exerts pressure on the walls of the blood vessels, much like water flowing through a hose. Our blood pressure naturally varies over the day, generally increasing when we are active and decreasing when we rest. However, if blood pressure remains high for extended periods of time it can lead to heart attacks, strokes and other serious health conditions.

In 2013, a new drug known as Adempas was approved to treat high blood pressure in the lungs. This drug helps a signaling molecule in the body called nitric oxide to activate an enzyme that widens blood vessels and in turn lower blood pressure. Previous studies have found that the enzyme – called soluble guanylate cyclase (sGC) – contains several distinct domains and that nitric oxide binds to a domain known as β H-NOX. However, it was not clear how β H-NOX and the other three domains fit together to make the three-dimensional structure of the enzyme, or how nitric oxide and Adempas activate it.

To address this question, Horst, Yokom et al. used a technique called cryo-electron microscopy to determine the three-dimensional structures of the inactive and active forms of a soluble guanylate cyclase from a moth known as *Manduca sexta*. To produce the active form of the enzyme, soluble guanylate cyclase was incubated with both nitric oxide and a molecule called YC-1 that works in similar way to Adempas. The structures revealed that nitric oxide and YC-1 caused β H-NOX and another domain to rotate by 71. This in turn caused the remaining two domains – known as the coiled-coil domains – to change shape, and all of these movements together led to the activated enzyme. The structures also revealed that YC-1 bound to a site on the enzyme between β H-NOX and the coiled-coil domains.

Understanding how a drug for a particular condition works makes it much easier to develop new drugs that are more effective at treating the same condition or are tailored to treat other diseases. Therefore, these findings will allow pharmaceutical companies and other organizations to develop new drugs for high blood pressure and other cardiovascular diseases in a much more precise way.
DOI: https://doi.org/10.7554/eLife.50634.002

structured assembly upon dimerization of sGC (*Campbell et al., 2014*). The CAT domains form a wreath-like structure with the active site at the dimer interface (*Derbyshire and Marletta, 2012*; *Hurley, 1998*).

Biochemical aspects of sGC activation by NO have been studied in great detail. Without a ligand bound to the β H-NOX domain, sGC has a low basal activity. A stoichiometric equivalent of NO relative to the sGC heterodimer results in the cleavage of the proximal histidine-iron bond and the formation of a distal five-coordinate ferrous nitrosyl enzyme with 15% of maximal activity (*Russwurm and Koesling, 2004*; *Cary et al., 2005*; *Fernhoff et al., 2009*; *Herzik et al., 2014*). This low-activity state of sGC will be referred to here as the 1-NO state; importantly, the $K_D$ of the ferrous nitrosyl heme is $1.2 \times 10^{-12}$ M thus in this state NO remains bound to the heme (*Zhao et al., 1999*). The activity of the 1-NO state can be increased to a maximally active state either by the addition of excess NO (xsNO), or by addition of small-molecule stimulators (*Stasch et al., 2011*). The benzylindazole compound YC-1 was the first reported small-molecule sGC stimulator (*Ko et al., 1994*). It was identified in a screen of compounds that inhibit platelet aggregation, one of the physiological responses of sGC activation. The molecular mechanisms by which NO and small molecule stimulator binding leads to enzyme activation remain unclear, despite the fact that the sGC-targeted drug Adempas discovered through a small molecule screen based on the YC-1 scaffold was approved by the FDA in 2013.

Visualization of the molecular steps in sGC activation has been a longstanding challenge. Crystal structures of individual truncated domains from various homologues of sGC have been reported (*Pellicena et al., 2004*; *Purohit et al., 2013*; *Ma et al., 2010*; *Winger et al., 2008*). Additionally, structures of related heterodimeric and multidomain proteins have provided insight into the higher order connectivity of sGC. The heterodimeric catalytic domain from *Homo sapiens* was solved in an inactive conformation (*Allerston et al., 2013*; *Seeger et al., 2014*). Structures of membrane-bound adenylate cyclases have been solved that contain catalytic domains as well as portions of or the

entire CC domains, helping to orient the C-terminal domains of sGC (*Vercellino et al., 2017*; *Qi et al., 2019*). However, the precise quaternary structural arrangement of domains in the N-terminal portion of sGC is not known. Consequently, the mechanism by which NO and small-molecule stimulators couple binding through the protein for activation are poorly understood.

In the absence of a high-resolution full-length structure, previous work has used alternative methods in attempts to understand interdomain interactions and how small molecules activate sGC. Crosslinking experiments and negative stain electron microscopy support the hypothesis that sGC is a flexible dumbbell-shaped particle, in which the CC domains serve to connect the H-NOX and PAS domains on one end of the dumbbell to the CAT domain on the other (*Campbell et al., 2014*; *Fritz et al., 2013*). Hydrogen deuterium exchange mass spectrometry (HDX-MS) has implicated the linker region between the PAS domains and the CC domains as critical in the activation mechanism, as both regions change in H/D exchange upon NO binding to sGC (*Underbakke et al., 2014*). Truncations of sGC have suggested that the β H-NOX domain directly inhibits the CAT domains (*Winger and Marletta, 2005*). Point mutagenesis was used to identify several residues thought to transmit the ligand occupancy of the gas-binding H-NOX domain to the catalytic domains, including β D102 and β D106 (*Baskaran et al., 2011a*; *Underbakke et al., 2013*). None of these methods, however, afford sufficient resolution to discern domain organization and interdomain communication for the full-length protein.

Here, we report the first full-length structures of sGC with and without activating ligands using cryoelectron microscopy (cryo-EM) to 5.1 and 5.8 Å resolution, respectively. We find that sGC adopts two dramatically different conformations, revealing a large-scale global conformational change in response to NO and YC-1. Density consistent with YC-1 is observed in the active state map directly between the β H-NOX and both CC domains. We also show the functional relationship of NO binding to sGC on overall organization of sGC in solution by small angle X-ray scattering (SAXS). By visualizing the quaternary structural changes that activate this NO receptor, we can now answer the long-standing question as to how the ligand binding in the regulatory domains of sGC is communicated to the catalytic domains.

## Results

### Characterization and activity of full-length *Manduca sexta* sGC

We selected *Manduca sexta* (*Ms*) sGC as a biochemically tractable protein which has 36% and 59% sequence identity when compared to the human homolog for the α and β monomers, respectively. A schematic of the domain organization is shown in *Figure 1a*. Additionally, the regulatory properties in response to NO and YC-1 mirror those of its human counterpart and YC-1 had been shown to increase the *Ms* sGC 1-NO state to maximal activity (*Hu et al., 2008*). While full-length *Ms* sGC has only been previously reported to be expressed in *E. coli*, we used a baculoviral expression protocol to produce reliable quantities of stable, full-length protein (*Figure 1—figure supplements 1* and *2*) (*Hu et al., 2008*). UV–visible absorption spectra of *Ms* sGC without ligands (unliganded, U), with xsNO, and with carbon monoxide (CO) show the expected ligand-dependent shifts of the Soret and Q bands, indicating the β H-NOX domain is competent to bind diatomic gases (*Figure 1b*).

We confirmed that *Ms* sGC displays catalytic activity similar to other well-characterized mammalian homologues, including *Homo sapiens* sGC. *Ms* sGC exhibits a low, basal activity in the unliganded state (71 ± 36 nmol/min/mg) and partial activation in the 1-NO state (309 ± 77 nmol/min/mg) (*Figure 1c*, *Figure 1—figure supplement 3*), consistent with previous reports of mammalian sGCs (*Fernhoff et al., 2009*). Maximal activity could be achieved by adding excess NO to the 1-NO sample (1988 ± 131 nmol/min/mg), or by the addition the sGC stimulator YC-1 to the 1-NO state (1522 ± 38 nmol/min/mg) (*Figure 1c*, *Figure 1—figure supplement 4*). Taken together, *Ms* sGC displays biochemical properties comparable to well-characterized mammalian sGCs.

### Inactive conformation of sGC exhibits bent coiled-coils

Cryo-EM was used to solve the structure of full-length inactive *Ms* sGC. Two-dimensional classification after removing poor particles showed intact density with two lobes with connective density between them similar to previously reported EM envelopes obtained by negative stain (*Figure 2—figure supplements 1–3*) (*Campbell et al., 2014*). Three-dimensional classification and refinement

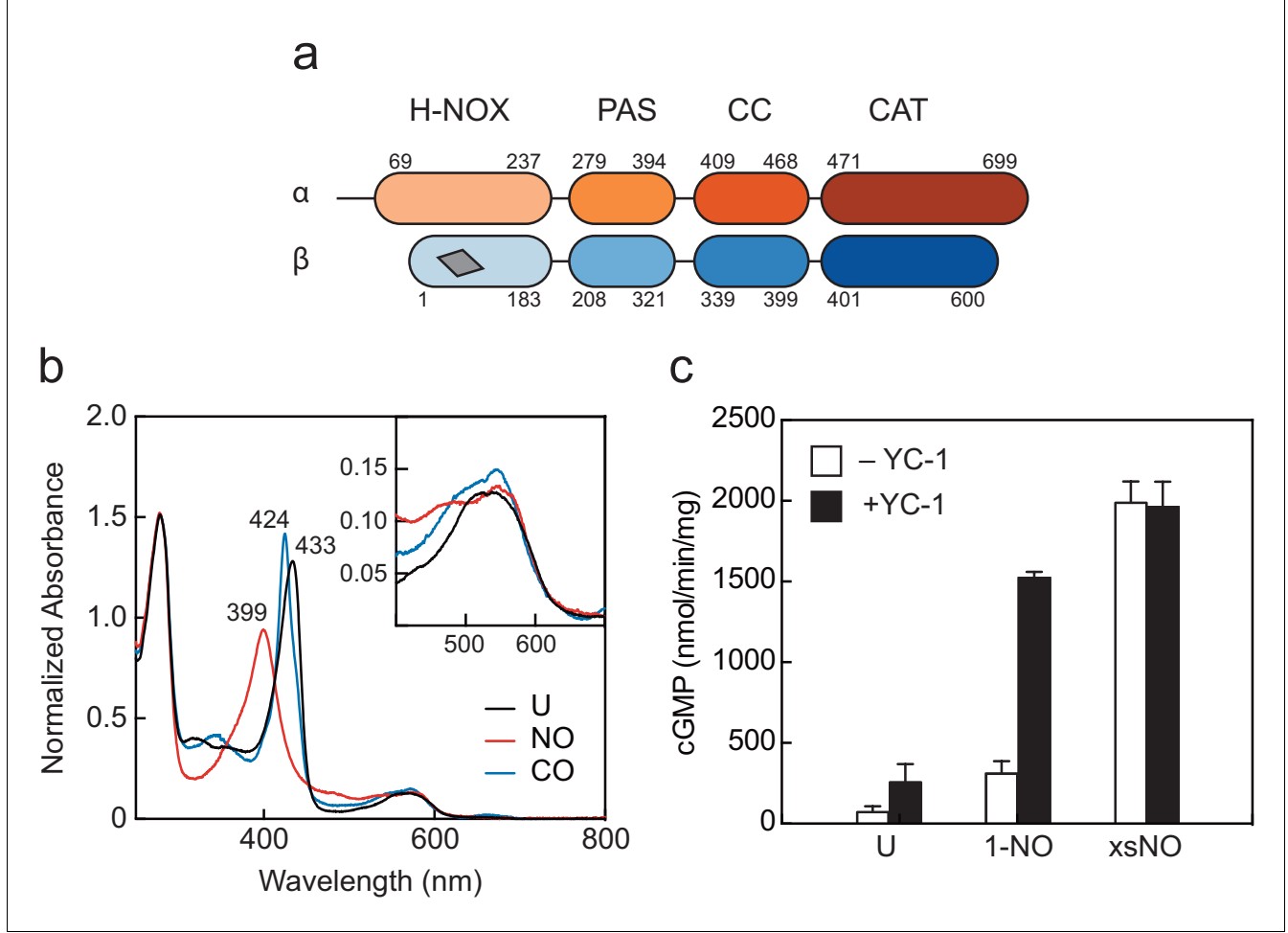

**Figure 1.** Domain arrangement and activation of *Manduca sexta* (*Ms*) soluble guanylyl cyclase (sGC). (a) Schematic representation of the *Ms* sGC heterodimer domain architecture. sGCs contain four domains: a heme nitric oxide/oxygen (H-NOX) binding domain, a Per/Arnt/Sim (PAS)-like domain, a coiled-coil (CC) domain, and a catalytic cyclase domain (CAT). The heme binding site in β H-NOX is represented by the gray quadrilateral. (b) UV–visible absorption spectra of *Ms* sGC in the unliganded (U), NO-bound, and CO-bound states. The wavelength maxima of the Soret peaks are indicated. Inset: Q bands show increased splitting upon gas binding. (c) Discontinuous cGMP activity assay for *Ms* sGC with various activation conditions: 1-NO, xsNO, and YC-1 ligands. Initial rates were taken from assays run at 25°C, pH 7.5 with 2 mM Mg•GTP as the substrate (see *Figure 1—figure supplement 1b*). cGMP formation was measured using an enzyme linked immunosorbent assay. The average initial rate is plotted, and the error bars reflect one standard deviation (n = 4).

DOI: https://doi.org/10.7554/eLife.50634.003

The following source data and figure supplements are available for figure 1:

**Source data 1.** Source Data for UV-Visible Absorption of Ms sGC.
DOI: https://doi.org/10.7554/eLife.50634.009

**Source data 2.** Source Data for Activity Assays for Ms sGC.
DOI: https://doi.org/10.7554/eLife.50634.010

**Figure supplement 1.** Representative SDS-PAGE gels for purification of *Ms* sGC.
DOI: https://doi.org/10.7554/eLife.50634.004

**Figure supplement 2.** Deconvoluted intact protein mass spectrum of purified *Ms* sGC.
DOI: https://doi.org/10.7554/eLife.50634.005

**Figure supplement 3.** UV–visible absorption spectra of *Ms* sGC used in activity assays.
DOI: https://doi.org/10.7554/eLife.50634.006

**Figure supplement 4.** Representative time courses of the cGMP assay for *Ms* sGC in different ligation states.
DOI: https://doi.org/10.7554/eLife.50634.007

**Figure supplement 5.** Topology diagram of β H-NOX secondary structure.
DOI: https://doi.org/10.7554/eLife.50634.008

yielded a single reconstruction with a 5.1 Å nominal resolution (*Figure 2—figure supplement 4*) that exhibited a clear dimer interface and distinct CCs (*Figure 2a*, *Supplementary file 1* - Table 1). While particles display a slight orientation preference, local resolution is uniform throughout the density map (*Figure 2—figure supplements 5* and *6*). Additionally, the overall map exhibits well-defined helices and continuous density. The two lobes of the structure are termed the 'regulatory' lobe and the 'catalytic' lobe, with the CC domains acting as a bridge between them (*Figure 2a*). The long axis of the regulatory lobe is positioned perpendicular to the catalytic lobe and is predicted to contain the H-NOX/PAS bundle, while the CAT domains are predicted to form the catalytic lobe (*Campbell et al., 2014*).

Several features enabled conclusive assignment of the α and β subunit domains in the density map, despite the intermediate resolution and pseudosymmetry between the two subunits (*Figure 2b*, *Video 1*). A rigid body global search of potential H-NOX positions revealed two possible positions for H-NOX domains within the regulatory lobe (*Pettersen et al., 2004*). The αF helix of the β H-NOX domain (*Figure 1—figure supplement 5*) binds a heme cofactor while the α H-NOX does not. Only the position closest to the catalytic lobe contains density for the heme cofactor (*Figure 2c*), hence this was assigned to the β subunit. The two CAT domains could be differentiated because the α CAT domain contains a C-terminal extension (residues 661–699) which is absent in the β CAT domain. Only one of the domains in the catalytic lobe contains extra C-terminal density and thus the CAT domains could be assigned unambiguously (*Figure 2—figure supplement 7*). Connective density from the CAT domains enabled clear differentiation of the two CC and PAS domains (*Figure 2d*, green). As a result, the α PAS is assigned as the top domain, while β is the bottom domain (*Figure 2b*). Homology models for *Ms* sGC domains were combined representing residues α 51–250, 279–699 and β1–183, 205–597 to create a full-length model (described in Materials and methods).

The CC domains are in a parallel orientation, and both domains have a clear bend, distinct from a previously determined CC X-ray structure (*Ma et al., 2010*). A linker extends from the C-terminus of the PAS domains and forms a rigid loop connecting the bent helix to the H-NOX/PAS domains (*Figure 2d*, light and dark green). The CC density bends sharply at residues α A422 and β L343. This bent region forms a buried helix, and along with the C-terminal PAS linker, creates an interaction nexus between the H-NOX, PAS, and CC domains (*Figure 2d*, highlighted in light and dark purple). The bent N-terminal portion of the CC domains (α 403–415 and β 333–345), is highly conserved across sGC homologues (*Figure 2—figure supplement 8*).

The CAT dimer displays a wreath-like fold with the monomers related by a twofold axis, typical for class III nucleotide cyclase domains (*Hurley, 1998*; *Zhang et al., 1997*). The CAT domains align well with a previous structure of an inactive guanylate cyclase (Cα RMSD of 1.3 Å to PDB ID: 4NI2), displaying an inaccessible nucleotide-binding pocket that would require a significant rearrangement for activation (*Figure 2—figure supplement 9*). In the absence of a substrate-bound structure of a guanylate cyclase, we compared our model to a substrate-bound adenylate cyclase. The alignment of the α chain of an active adenylate cyclase structure (PDB ID: 1CJK, gray) with the α CAT domain of our inactive *Ms* sGC structure reveals that the Cα of β N538 from the β CAT domain is within 2 Å of the bound nucleotide analog in the adenylate cyclase structure (*Figure 2e*). This state is thus in a closed conformation that is sterically incompatible with nucleotide binding, explaining the lack of guanylate cyclase activity.

## Activated sGC extends the regulatory lobe from catalytic core

To elucidate conformational changes associated with sGC activation, a structure of sGC bound to NO and the small molecule stimulator YC-1 was determined. We elected to supplement the xsNO *Ms* sGC with YC-1 to generate the most stable activated conformation. Particles of active *Ms* sGC were well-dispersed in vitreous ice (*Figure 3—figure supplements 1* and *2*). Cleaned two-dimensional class averages exhibit a two-lobed density that is distinct from the inactive structure (*Figure 3—figure supplement 3*). The final 3D reconstruction of the active state at 5.8 Å resolution displays clear density for the CCs, β H-NOX, and α helices throughout the structure (*Figure 3a*, *Video 2*, *Figure 3—figure supplements 4–6*). Only diffuse density was present for the α H-NOX, likely due to increased flexibility of this domain and it was thus omitted in the model (*Figure 3b*).

The orientation of the domains in the regulatory lobe is maintained from the inactive state to the active state, and the β H-NOX position is readily identified via the bound heme cofactor density

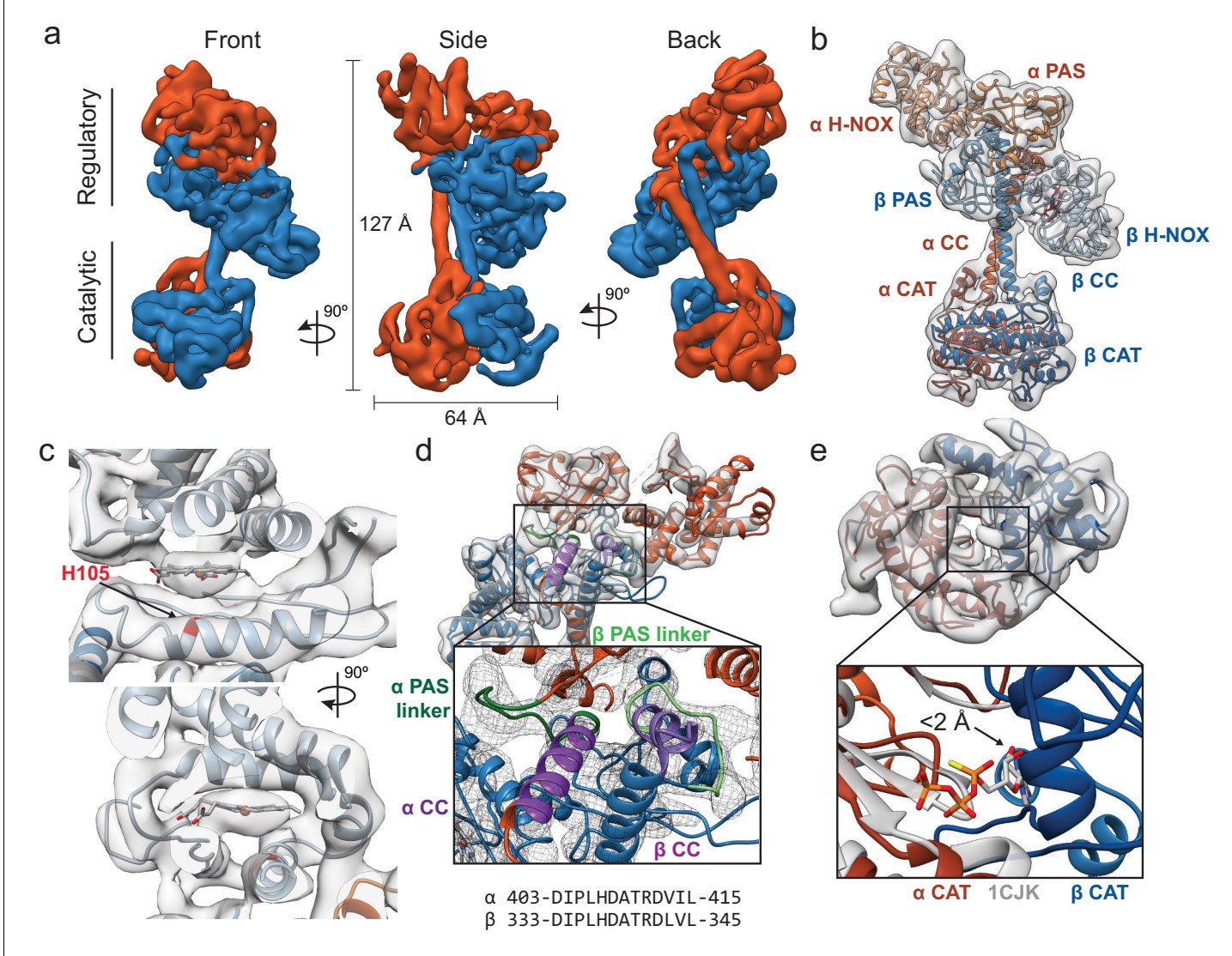

**Figure 2.** Inactive *Ms* sGC forms a bent coiled-coil structure. (a) Views of cryo-EM density for the inactive state, colored by subunit (α - orange, β - blue). Dimensions of the complex are shown in black. The two lobes of the enzyme are denoted 'regulatory' and 'catalytic'. (b) Molecular model of *Ms* sGC with domains labeled and colored as in *Figure 1a*. The heme is colored gray. (c) Two views of the β H-NOX domain with heme shown in gray and H105 in red (d) View of the bent coiled-coil (purple) and PAS linker (green) shown in shades for the α (dark) and β (light) dimer. Closeup shows connective density between the PAS and CC domains (threshold 12σ). The sequences of the bent CC domains are shown below. (e) View of the inactive catalytic dimer fit into density (α - orange, β - blue). Close up shows aligned active adenylyl cyclase (α CAT domain) and nucleotide in gray (PDB: 1CJK) compared with the inactive sGC model. The distance between the substrate analogue and nearest backbone Cα (β N538) is shown.

DOI: https://doi.org/10.7554/eLife.50634.011

The following figure supplements are available for figure 2:

**Figure supplement 1.** Representative micrograph with inlet of corresponding FFT.
DOI: https://doi.org/10.7554/eLife.50634.012

**Figure supplement 2.** Electron microscopy flowchart for processing of inactive.
DOI: https://doi.org/10.7554/eLife.50634.013

**Figure supplement 3.** Selected 2D class averages.
DOI: https://doi.org/10.7554/eLife.50634.014

**Figure supplement 4.** FSC curve from cryosparc2 with gold standard 0.143 FSC shown in blue.
DOI: https://doi.org/10.7554/eLife.50634.015

**Figure supplement 5.** Plot of single particle population based on orientation parameters.
DOI: https://doi.org/10.7554/eLife.50634.016

**Figure supplement 6.** Inactive reconstruction colored by local resolution calculated using cryosaprc2 implementation of blocres.
*Figure 2 continued on next page*

*Figure 2 continued*

DOI: https://doi.org/10.7554/eLife.50634.017

**Figure supplement 7.** Closeup view of the modeled inactive.

DOI: https://doi.org/10.7554/eLife.50634.018

**Figure supplement 8.** The coiled-coil domain of eight sequences from four heterodimeric biochemically verified NO-responsive sGCs and five sequences from atypical sGCs that sense O₂ are aligned, with conserved residues are highlighted in blue.

DOI: https://doi.org/10.7554/eLife.50634.020

**Figure supplement 9.** Overlay of the PDB: 4NI2 (gray) and the inactive CAT dimer (colored by α and β).

DOI: https://doi.org/10.7554/eLife.50634.019

**Figure supplement 10.** Inactive model of *Ms* sGC is shown with the bent helix in purple and the S-helix in red.

DOI: https://doi.org/10.7554/eLife.50634.021

(*Figure 3c*). A triangular-shaped lobe of unassigned density is present in the active state map but not in the inactive state map (*Figure 3c*, arrow). This density is located directly between the β H-NOX domain and the α and β CC domains, adjacent to either the A or B pyrrole of the heme (opposite of the propionate chains). The volume likely corresponds to YC-1 (304.3 Da) and with the current resolution two orientations of this small molecule are possible (*Figure 3c*). Residues within 5 Å of the density include α CC 422–427 (AQDGLR), and β V39, F77, C78, Y112, and Q349.

Although the density for the β H-NOX and PAS domains is similar between the inactive and active states, the overall shape of the molecule is more linear (*Figure 3b*). Continuous, linear helices from α L406–L457 and β A333–Y386 are seen in the active state (*Figure 3d*, highlighted in purple). This conformation is more aligned with predicted CC length and is more similar to the previously published crystal structures in terms of the range of residues seen in an unbent coiled-coil (*Ma et al., 2010*; *Qi et al., 2019*).

The CAT heterodimer in the active state still forms a wreath-like geometry, but with the two monomers moved apart from one another roughly perpendicular to the CC domain axis (*Figure 3e*). An alignment of the α chain of an active adenylate cyclase structure (PDB ID: 1CJK, gray) with the α CAT domain of our active *Ms* sGC structure shows the Cα of β N538 is now greater than 4 Å from the bound nucleotide analog, providing sufficient space for the nucleotide to bind in the active site. The *Ms* sGC CAT domain thus adopts an open conformation with the apparent capacity to bind substrate.

## Allosteric conformational rearrangement of sGC

Binding of NO to β H-NOX initiates the signal transduction event, thus interfaces were examined between the inactive and active states. A significant rearrangement of the quaternary structure of *Ms* sGC occurs upon activation with the regulatory domain rotating 71° and the catalytic domain rotating 40° (*Figure 4a*, *Video 3*). In both inactive and active states, the β H-NOX domain is the closest domain of the regulatory lobe to the CAT domains (*Figure 4a*); however, in neither model does the β H-NOX domain come into direct contact with the CAT domains. In the inactive state, the Cα of β I47 is 12 Å away from the Cα of β H399 (*Figure 4a*, gray), too far for direct contact. The closest Cα-Cα distance of residues between the β H-NOX domain and the CAT domains in the active state is between β M33 and α W468 at 21 Å (*Figure 4a*, black), remaining too far for direct contact between the β H-NOX and the CAT dimer. Therefore, allosteric communication is

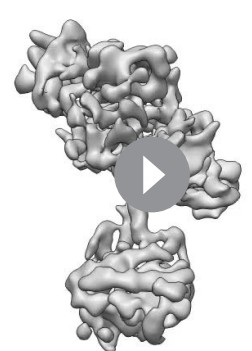

Inactive

**Video 1.** Cryo EM Reconstruction of the Inactive sGC Conformation.

DOI: https://doi.org/10.7554/eLife.50634.022

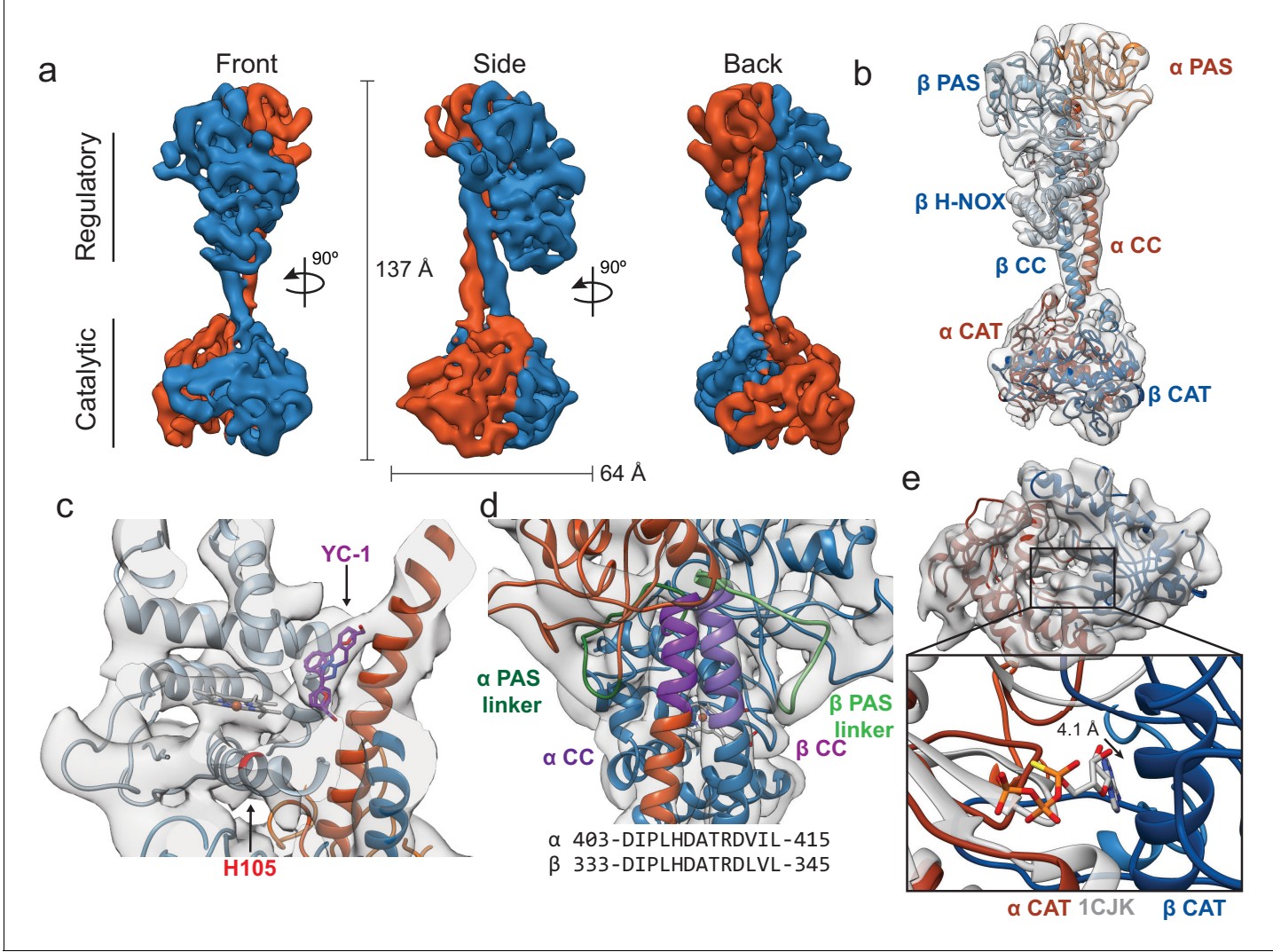

**Figure 3.** Active *Ms* sGC forms an elongated structure. (**a**) Views of cryo-EM density for the active state, colored by subunit (α - orange, β - blue). Dimensions of the complex are shown in black. (**b**) Molecular model of *Ms* sGC colored as in *Figure 1a*. The heme is colored gray. The α H-NOX domain is not shown due to lack of density. (**c**) View of the β H-NOX domain with heme shown in gray and β H105 in red. Two fits for the stimulator, YC-1, are shown in purple (**d**) Model of the extended coiled-coil region (purple), highlighting the PAS linker region (green) shown in shades for the α (dark) and β (light) dime. The sequences of the bent CC region are shown below. (**e**) View of active catalytic dimer (α - orange, β - blue) shown fit into the reconstruction density. Closeup shows aligned active adenylate cyclase structure (PDB: 1CJK) with bound nucleotide (gray). The distance between the substrate analogue and Cα of the β N538 is shown.

DOI: https://doi.org/10.7554/eLife.50634.023

The following figure supplements are available for figure 3:

**Figure supplement 1.** Representative micrograph with inlet of corresponding FFT.

DOI: https://doi.org/10.7554/eLife.50634.024

**Figure supplement 2.** Electron microscopy flowchart for processing of active sGC data.

DOI: https://doi.org/10.7554/eLife.50634.025

**Figure supplement 3.** Selected 2D class averages.

DOI: https://doi.org/10.7554/eLife.50634.026

**Figure supplement 4.** FSC curve from cryosparc2 with gold standard 0.143 FSC shown in blue.

DOI: https://doi.org/10.7554/eLife.50634.027

**Figure supplement 5.** Plot of single particle population based on orientation parameters.

DOI: https://doi.org/10.7554/eLife.50634.028

**Figure supplement 6.** Active reconstruction colored by local resolution calculated using cryosaprc2 implementation of blocres.

DOI: https://doi.org/10.7554/eLife.50634.029

*Figure 3 continued on next page*

*Figure 3 continued*

**Figure supplement 7.** Differential H/D Exchange values representing change in percent D by subtracting the active state from the inactive state.
DOI: https://doi.org/10.7554/eLife.50634.030

transmitted through changes in CC positioning, not through direct H-NOX:CAT interactions.

Interaction surfaces, defined by a $\leq 10$ Å Cα to Cα distance (*Donald et al., 2011*), occur between the β H-NOX and the PAS or CC region in both inactive and active states (*Figure 4b,c*). In the inactive state, the αE, αF, and β2 secondary structural elements of the β H-NOX (82–126, green) form an extensive interface with the β PAS (270–275, red) and the α PAS-CC linker (400–419, blue) regions (*Figure 4b*, *Figure 1—figure supplement 5*). Overall, these interfaces are maintained in the active state (*Figure 4c*), in spite of the straightening by the CC domains. Activation of sGC induces the formation of an interface between β CC (355-367) and β H-NOX (33-41) (*Figure 4c*, purple). These residues on the β H-NOX comprise a loop located between helix αB and αC in the β H-NOX domain which was extended away from the CC in the inactive state. The large conformational change of the H-NOX/PAS bundle is necessary for the formation of the new interface in the active site, and thus this interaction could be critical for stabilizing the active sGC conformation.

In the active structure, the CCs undergo a significant conformational change and rotation relative to the CAT dimer (*Figure 4d*). The active state CC domains are completely extended (56 Å and 63 Å for αCC and βCC respectively in the inactive state, to 74 Å for both CC in the active state), as the bend present in the inactive state moves away from the CAT domains to be in line with the C-terminal portion of both CC domains. The CC conformational change upon activation twists the CC interface such that the β CC helix rotates relative to the α CC by 72 degrees (*Figure 4d*). The dramatic rearrangements of the H-NOX/PAS bundle leads to unbending of the CC domains, a change in their orientations as they project from the regulatory lobe, and finally to opening of the nucleotide-binding pocket (*Figure 4e*). Alignment to the α CAT domain shows a clear rotation of the β CAT domain away from α CAT domain by 40° (*Figure 4e*, left). This rotation entails the pinching together of the base of the CC domains (*Figure 4e*, middle). In combination, these motions open the GTP-binding pocket by moving the β CAT active site helix (535-545) away from the α CAT domain and into an active conformation.

## SAXS shows a distribution of inactive and active states in solution

Small angle X-ray scattering (SAXS) was used to interrogate conformational changes with and without NO in solution. Experiemnts with sGC stimulators were not included as the limited solubility of YC-1 is precludes the collection of scattering data. Size exclusion chromatography (SEC)-SAXS and SEC-multi angle light scattering (MALS) chromatograms show a symmetrical peak for both samples with little variation in radius of gyration ($R_g$) and estimated MW across the peaks, indicating sample homogeneity (*Figure 5—figure supplement 1*).

The UV-visible absorption of the Soret band of the heme and of the protein itself were used to evaluate the ligation state of *Ms* sGC in the SAXS experiment. The ratios of Soret band to protein peak in the full UV-visible absorption spectra are 0.85, 0.35, and 0.20 in the unliganded, 1-NO, and xsNO states, respectively (*Figure 1b*, *Figure 5—figure supplement 1*). The ratios of these two peaks in the SEC-SAXS experiments were 0.77 with NO and 0.32 without NO (*Figure 5—figure supplement 1*). The 432/280 ratio in the SAXS sample without NO is less than in the full UV-Visible spectra, indicating that the sample

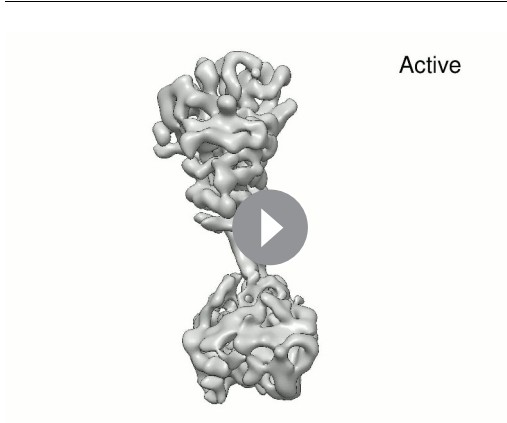

Active

**Video 2.** Cryo EM Reconstruction of the Active sGC Confromation.
DOI: https://doi.org/10.7554/eLife.50634.031

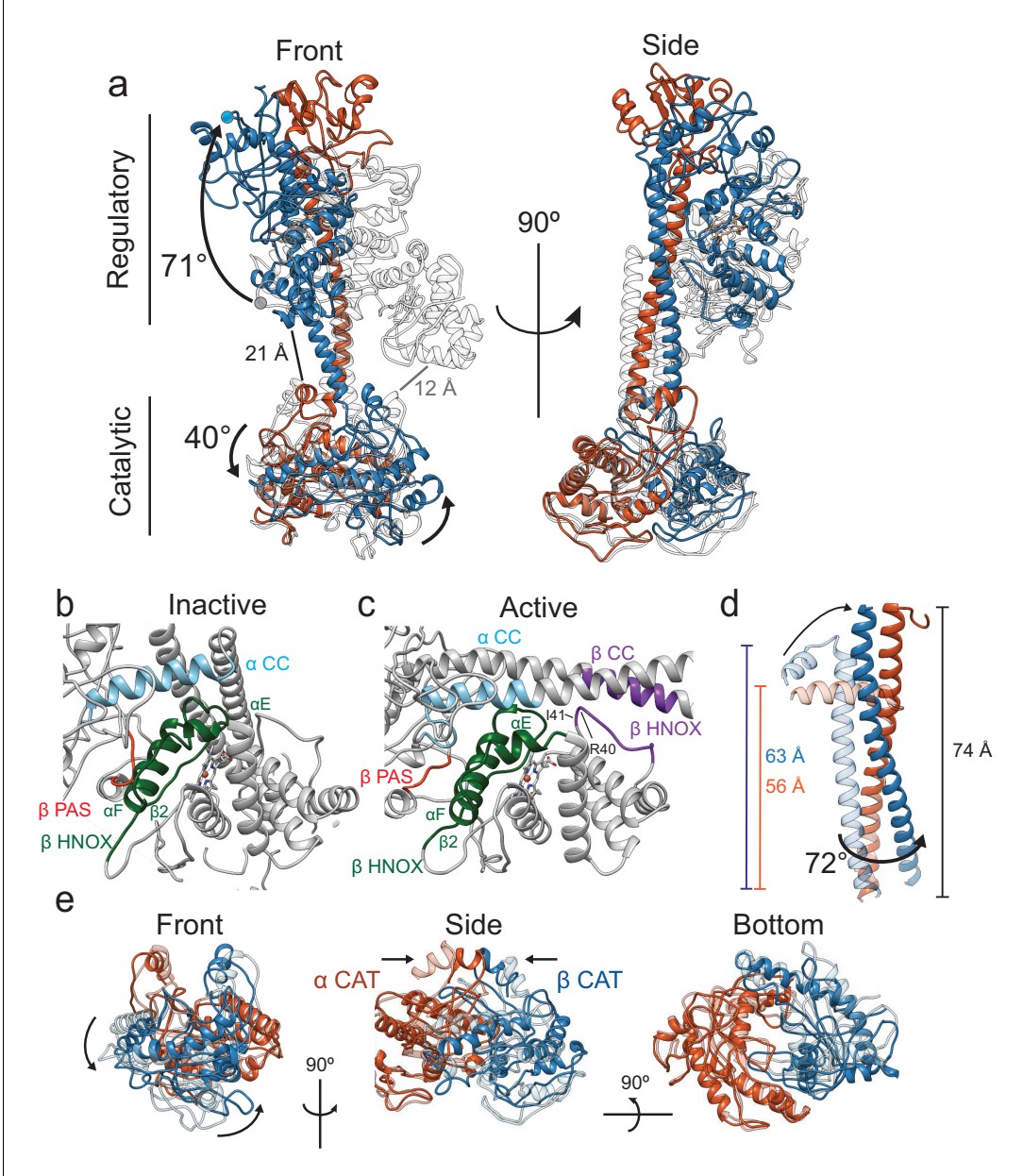

**Figure 4.** Conformational rearrangements of *Ms* sGC upon activation. (**a**) Overlay of the inactive (transparent, gray) and active (α - orange, β – blue) shown in two views. Rotation of the regulatory domain and CAT dimer are shown with arrows and label with degree of rotation. Distances between the β H-NOX domain and CAT dimer are labeled for inactive (gray) and active (black). (**b**) Close up view of the inactive interfaces between the β HNOX (green), β PAS (red), and α CC (blue). (**c**) Closeup view of the active interfaces colored as in *Figure 4b* with the β H-NOX:β CC interface in purple. (**d**) Overlay of the inactive (transparent) and active (colored) CC domains when aligned to the active α CC domain. Dimensions of the CC and rotation are labeled in color for the inactive and black for the active. (**e**) Three views of aligned inactive (transparent) and active (colored) CAT dimers are shown. Direction of rotation (curved arrows) and pinching (arrows) are shown.

DOI: https://doi.org/10.7554/eLife.50634.032

had a slightly lower heme incorporation. The ratio for the SAXS sample with NO was lower than that expected for 1-NO, but much larger than predicted for the xsNO ratio, indicating that the SAXS samples were in the unliganded and 1-NO state during elution, respectively.

Inactive State

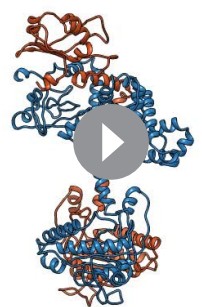

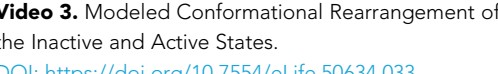

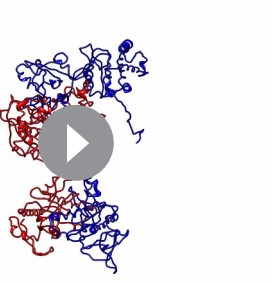

**Video 4.** Example of conformational sampling produced by BILBOMD (*Pelikan et al., 2009*).
DOI: https://doi.org/10.7554/eLife.50634.042

**Video 3.** Modeled Conformational Rearrangement of the Inactive and Active States.
DOI: https://doi.org/10.7554/eLife.50634.033

The $R_g$ for the unliganded and 1-NO state were 43.1 ± 0.4 and 43.8 ± 0.2 Å, respectively, representing an elongation of the structure (*Figure 5—figure supplement 1*, *Supplementary file 2* - Table S2). Additionally, the Pair-distance function (P(r)), which shows a composite of the inter-atomic distances, displays a distinct shift between the unliganded and 1-NO states in the region from ~70 to~100 Å. This corresponds to an extension of the regulatory lobe from catalytic lobe (*Figure 5a*). Shifts in the secondary peaks of the Kratky plots (*Figure 5—figure supplement 2*) further suggest separation of the regulatory and catalytic lobes upon activation.

Using the inactive model obtained from cryo-EM as an initial Cα model, a rigid-body modeling pipeline was developed to systematically explore conformational space (see Online Methods) (*Pelikan et al., 2009*). A minimal ensemble search was performed over thousands of sGC conformations and corresponding scattering curves were calculated and compared to the experimental data. The result of this minimal ensemble search confirmed the presence of a single sGC conformation while in an inactive state (*Figure 5b*, *Figure 5—figure supplement 3*), which overlays with the inactive structure with an RMSD of 2.0 Å. The 1-NO state was best modeled as an ensemble of two states, with the majority (72%) of the sample consistent with the inactive model (*Figure 5b*). However, the remainder of the sample (28%) is consistent with a more elongated conformation, one that is in between the inactive and active state obtained by cryo-EM. The Cα RMSD of the partially active SAXS model and the active state cryo-EM model is 13.5 Å. This analysis shows that sGC adopts a mixture of inactive and active conformations in the 1-NO state, suggesting that NO binding at the heme establishes an equilibrium between these two conformational extremes (*Figure 5—figure supplement 3*).

## Discussion

The structures of the inactive and active states of full-length sGC provide detailed insight into the molecular mechanism of activation of *Ms* sGC. The most unexpected feature of the inactive state of the enzyme are the bends in the CC domains. Straightening of the bent CCs is the clearest structural link between the regulatory and catalytic lobes during activation. Previous HDX-MS data showed a differential exchange pattern for the CC domains, where the N-terminal portion of the α CC and the C-terminal portion of the β CC increase in H/D exchange upon NO binding, while the C-terminal portion of the α CC and the N-terminal portion of the β CC decrease in H/D exchange (*Figure 3—figure supplement 7*) (*Underbakke et al., 2014*). The changes seen in HDX in the CCs near the bend are thus consistent with the large-scale rearrangements of the CCs that we have now observed in the cryo-EM. Furthermore, the bent portion of the CC domain is highly conserved among sGC homologues, consistent with its centrality to allosteric communication.

The C-terminal portions of the CC domains undergo a 72° twist and contain a motif known as the signaling helix (or S-helix) (*Anantharaman et al., 2006*). Similar to sGC, proteins with this motif contain both receptor and output domains that are connected by dimeric coiled-coils. The inactive and active sGC structures are the first full-length enzyme with a S-helix motif to be characterized. This

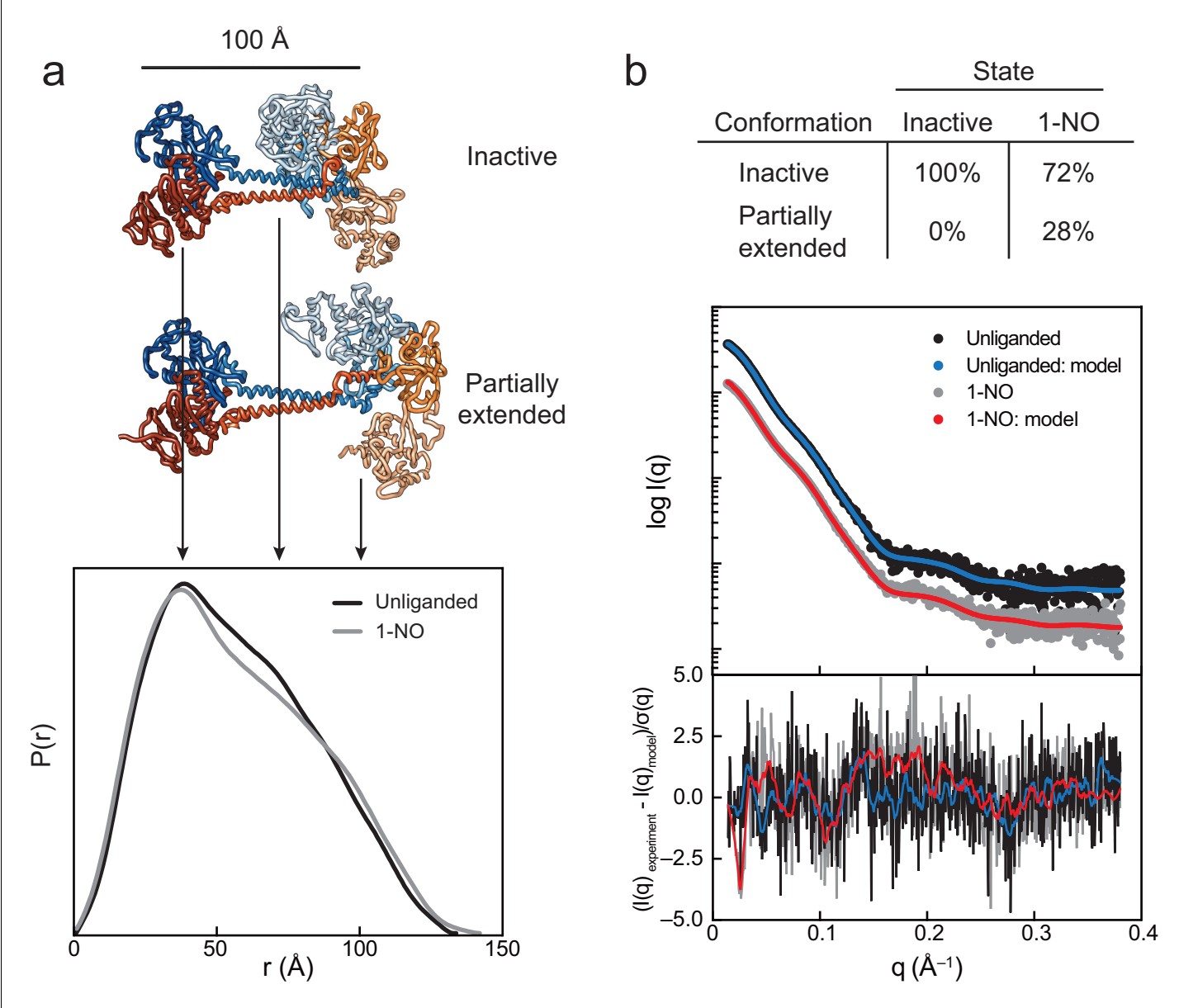

**Figure 5.** SAXS analysis of the activation of sGC. (**a**) Normalized P(r) for the unliganded (black) and 1-NO (gray) states of sGC are shown together with corresponding models of the inactive and partially extended conformations of sGC (N-terminal and C-terminal tails are omitted for clarity). The area of each P(r) is normalized relative to the SAXS-determined molecular weights (***Supplementary file 2*** - Table 2). (**b**) Experimental (black and gray) and theoretical (blue and red) SAXS profiles for the solution state models in the panel a. SAXS fits are shown together with the fit residuals in the below graph.

DOI: https://doi.org/10.7554/eLife.50634.034

The following source data and figure supplements are available for figure 5:

**Source data 1.** Source Data for Pair Distribution plots of Ms sGC.
DOI: https://doi.org/10.7554/eLife.50634.038
**Source data 2.** Source Data for experimental and modeled scattering of Ms sGC.
DOI: https://doi.org/10.7554/eLife.50634.039
**Figure supplement 1.** Top: Small angle X-Ray scattering chromatograms with intensity (lines) and $R_g$ (symbols) values for each frame merged across the SEC peak.
DOI: https://doi.org/10.7554/eLife.50634.035
**Figure supplement 2.** Normalized Kratky plot for the inactive (black) and active state (gray) of sGC.
DOI: https://doi.org/10.7554/eLife.50634.036

*Figure 5 continued on next page*

*Figure 5 continued*

**Figure supplement 3.** Top: Best fit conformations for the inactive state of sGC.

DOI: https://doi.org/10.7554/eLife.50634.037

motif aligns with C-terminal part of the CC domain, part of the 72° twist described above (*Figure 2—figure supplement 10*). It is possible that the twisting motion of these domains is a more general mechanism of activation for proteins with S-helices.

In transitioning between the inactive and active states, the H-NOX/PAS bundle rotates 71° in order for β H-NOX residues 33–41 to form an interface with residues β 355–367 of the β CC domain (*Figure 4c*, purple). The αH helix of the β H-NOX domain contains the proximal histidine that binds to the heme cofactor (*Zhao et al., 1998*). Crystal structures of bacterial H-NOX domains with and without NO show a rotation in the αF helix of the β H-NOX. However, in our full-length structures, the αF of the β H-NOX retains its interface with the β PAS and the α CC domain in both states (*Figure 4b and c*). More strikingly, binding of YC-1 at a site bridging the β H-NOX and CC domain highlights how stabilization of this contact drives CC unbending (*Figure 3c*). This interface corroborates specific point mutations known to decrease activity of the full-length protein; specifically, single point mutants of β I41E in the β H-NOX domain retain basal activity but are only minimally activate with excess NO, implying that these sGC variants are catalytically competent but not as sensitive to stimulation as the wildtype enzyme (*Underbakke et al., 2013*; *Baskaran et al., 2011b*). The structural finding that residue Ile41 of the β H-NOX form contacts with the unbent β CC that are specific for the active conformation explains these phenotypes. Destabilizing these contacts by mutating Ile41 thus blocks this interface and prevents sGC from reaching the active conformation. While this paper was under review, a cryo-EM map of activated *H. sapiens* sGC was reported, where the activation was achieved by using xsNO (*Kang et al., 2019*). This xsNO map and the xsNO + YC-1 map reported here overlay well, except for the proposed density for YC-1. This lends support to our assignment of the YC-1 binding site at the β H-NOX- β CC interface. These insights suggest that the β H-NOX- β CC contact may be the most critical allosteric switch in the regulatory lobe of sGC.

While conformational changes are readily observed by comparing the two cryo-EM structures, the SAXS implies that the 1-NO state can sample similar conformations (*Figure 5a*). Although only a fraction of the population exhibits an extension, inspection of the Soret to protein UV-visible absorption indicates that the protein is in the 1-NO state. These data support the hypothesis that when the first NO binds to the heme of the β H-NOX, sGC adopts an equilibrium between the inactive and a partially extended conformation, with a $K_{eq}$ = [active]/[inactive] = [0.72]/[0.28]=0.39. This is corroborated by the activity data from the 1-NO state, which exhibits 15% of the maximal activity. The conformational heterogeneity observed in SAXS analysis could represent the physiological state of sGC at basal cellular conditions.

In the absence of increased NO concentrations, sGC stimulators can maximally activate cGMP production and are now used to treat forms of pulmonary hypertension (*Koglin et al., 2002*; *Follmann et al., 2013*). YC-1 was present in the sample used to generate the active state reconstruction of *Ms* sGC, and extra density is seen near the new β H-NOX: β CC interface (*Figure 3c*). Distinct changes in both resonance Raman and electron paramagnetic resonance spectroscopy signatures for heme ligands have been detected with both YC-1 and BAY 41–2272 supporting this proposed binding site (*Derbyshire, 2008*). Previously, a stimulator binding site was proposed between helices αA and αD in the β H-NOX domain based on cross linking and NMR data (*Wales et al., 2018*). However, there is no density for YC-1 present in the active state reconstruction between αA and αD helices. We note that our study used YC-1 compared to IWP-854, IWP-051 and BAY 41–2272 previously used, which could explain the difference in binding sites. Visualization of YC-1 bound to sGC is an important development as a proof of concept that small molecule sGC stimulators can now be characterized structurally.

Adenylyl cyclases (AC) catalyze a similar reaction mechanism as sGC. Mammalian membrane-associated ACs contain two catalytic domains which form the intrapolypeptide equivalent of a heterodimer. The catalytic domains in active mammalian AC heterodimer rotate by 7° with respect to an inactive homodimeric counterpart bound to two molecules of forskolin (*Zhang et al., 1997*;

*Tesmer et al., 1997*; *Hurley, 1999*). We observe a rotation of the sGC CAT domains about the same axis, although the magnitude is larger for sGC, at 40°. A full-length membrane bound AC (AC-9) was recently solved in the active nucleotide-bound state (*Qi et al., 2019*). The CC domains between the full-length active sGC and AC9 structures overlay well, consistent with a common active geometry for GCs and ACs. The AC9 has a very different angle between the CC and CAT domains. The rotation of the sGC regulatory lobe is sterically incompatible with the presence of a membrane, thus by their different natures as soluble and membrane-associated enzymes, sGC and ACs must differ in the detailed modes of allosteric communication.

To date, a detailed understanding of sGC activation has been hampered by the lack of full-length structures. The inactive and active structures in tandem with established studies lead to formation of new hypotheses for the activation of sGC. First, activity assays with sGC truncations suggested that direct interaction of the β H-NOX domain and the CAT dimer was responsible for sGC inhibition (*Winger and Marletta, 2005*). However, the structural data shows no direct interaction between the β H-NOX and the CAT domain in either conformation. Instead, the formation of the new β H-NOX/β CC interface in the active state stimulates the active CAT conformation. Activation leads to a global conformational rearrangement of the heterodimer, elongating the structure; however, only a single equivalent of NO is required to cleave the bond between the heme Fe center and the ligating histidine residue. Maximal activation of sGC involves either the binding of a second NO molecule to a non-heme site on sGC or a small molecule stimulator stabilizing the active state (*Guo et al., 2017*; *Horst and Marletta, 2018*).

*Figure 6* depicts the proposed physiological activation sequence of sGC, where unliganded sGC adopts a more compact conformation with bent CC domains. Upon NO binding to the heme, an equilibrium of conformational states is established, with the partially elongated state affording about 15% of the maximal activity. Given the very tight association between NO and the heme, this represents the basal cellular activity. Finally, in the presence of either an increase in NO concentration or sGC stimulating molecules, the completely extended, fully active state is reached and sGC reaches maximal catalytic activity. Using these structures as a starting point, new avenues of exploration can now be undertaken, to elucidate the molecular mechanisms of excess NO activation. Critical residues for interdomain interactions along with the proposed stimulator binding site can be characterized in more detail. Having resolved the two structural states of an important therapeutic target, the structures of sGC will influence rational design of improved drugs for diseases associated with NO signaling impairment.

# Materials and methods

## Key resources table

| Reagent type (species) or resource | Designation | Source or reference | Identifiers | Additional information |
|---|---|---|---|---|
| Gene (*Manduca sexta*) | *Ms* sGC α1 | Integrated DNA Technologies | Genbank: AF062750 | Uniprot: O77105 |
| Gene (*Manduca sexta*) | *Ms* sGC β1 | Integrated DNA Technologies | Genbank: AF062751 | Uniprot: O77106 |
| Strain, strain background (*E. coli*) | XL1-Blue | UC Berkeley MacroLab | | Cloning strain |
| Strain, strain background (*E. coli*) | DH10-Bac | UC Berkeley MacroLab | | Transposition Strain |
| Recombinant DNA reagent | pFastBac (plasmid) | Thermo Fisher | | Donor Plasmid |

*Continued on next page*

*Continued*

| Reagent type (species) or resource | Designation | Source or reference | Identifiers | Additional information |
|---|---|---|---|---|
| Recombinant DNA reagent | pFastBac_Ms_α1_His6 | This paper | | See Materials and methods section Construction of Plasmids |
| Recombinant DNA reagent | pFastBac_Ms_β1 | This paper | | See Materials and methods section Construction of Plasmid |
| Sequence-based regent | Primers | This paper | | See primer table in Materials and methods |
| Cell line (*Spodoptera frugiperda*) | SF9 cells | UC Berkeley Cell Culture Facility | RRID: CVCL_0549 | |
| Commercial Assay or Kit | Cellfectin II | Thermo Fisher | | Transfection reagent |
| Transfected Construct (*Manduca sexta*) | bMON14272_Ms_α1_His6 (bacmid) | This paper | | Transfected construct |
| Transfected Construct (*Manduca sexta*) | bMON14272_Ms_β1_His6 | This paper | | Transfected construct |
| Chemical compound, drug | DEA NO NOate | Cayman Chemical | Item No: 82100 | CAS: 372965-00-9 |
| Chemical compound, drug | PROLI N ONOate | Cayman Chemical | Item No: 82145 | CAS: 17894 8-42-0 |
| Commercial Assay or Kit | cGMP ELISA | Enzo Life Science | ADI-901–013 | |

## Materials

All chemicals were purchased from commercial vendors and used without further purification, unless otherwise noted. Primers and gBlocks were obtained from Integrated DNA Technologies (Coralville, IA). Gibson Master Mix was purchased from New England Biolabs (Ipswich, MA). PrimeSTAR Max DNA polymerase and TALON superflow resin was purchased from Takara Bio USA (Mountain View, CA). DNA purification kits were purchased from Qiagen (Germantown, MD). ExCell-420 media, sodium dithionite ($Na_2S_2O_4$), DNAse I from bovine pancreas, β-mercaptoethanol, sodium phosphate, dibasic ($Na_2HPO_4$), sodium carbonate ($Na_2CO_3$), potassium phosphate ($K_2HPO_4$), guanosine 5′-triphosphate sodium salt (GTP) (>95%, HPLC), and zinc acetate ($Zn(CH_3CO_2)_2$) were purchased from Millipore Sigma (Burlington, MA). 4-(2-aminoethyl)benzenesulfonyl fluoride hydrochloride (AEBSF) was purchased from Research Products International (Mount Prospect, IL). Isopropyl β-D-1-thiogalactopyranoside (IPTG), 4-(2-hydroxyethyl)piperazine-1-ethanesulfonic acid (HEPES), and the Pierce BCA Protein Assay Kit were purchased from Thermo Fisher Scientific (Waltham, MA). Benzamidine, sodium chloride (NaCl), magnesium chloride ($MgCl_2$), and glycerol were purchased from VWR Life Science (Visalia, CA). Dithiothreitol (DTT) was purchased from Bachem (Bubendorf, Switzerland). Imidazole was purchased from Oakwood Chemical (West Columbia, SC). Vivaspin spin concentrators were purchased from Sartorius (Concord, CA). BioSpin six desalting columns were purchased from BioRad (Hercules, CA). Diethylamine NONOate (DEA NONOate), PROLI NONOate, diethylamintriamine NONOate (DETA NONOate), and YC-1 were purchased from Cayman Chemical Company (Ann Arbor, MI). Carbon monoxide (CO,>99%) gas was purchased from Praxair Inc (Danbury, CT).

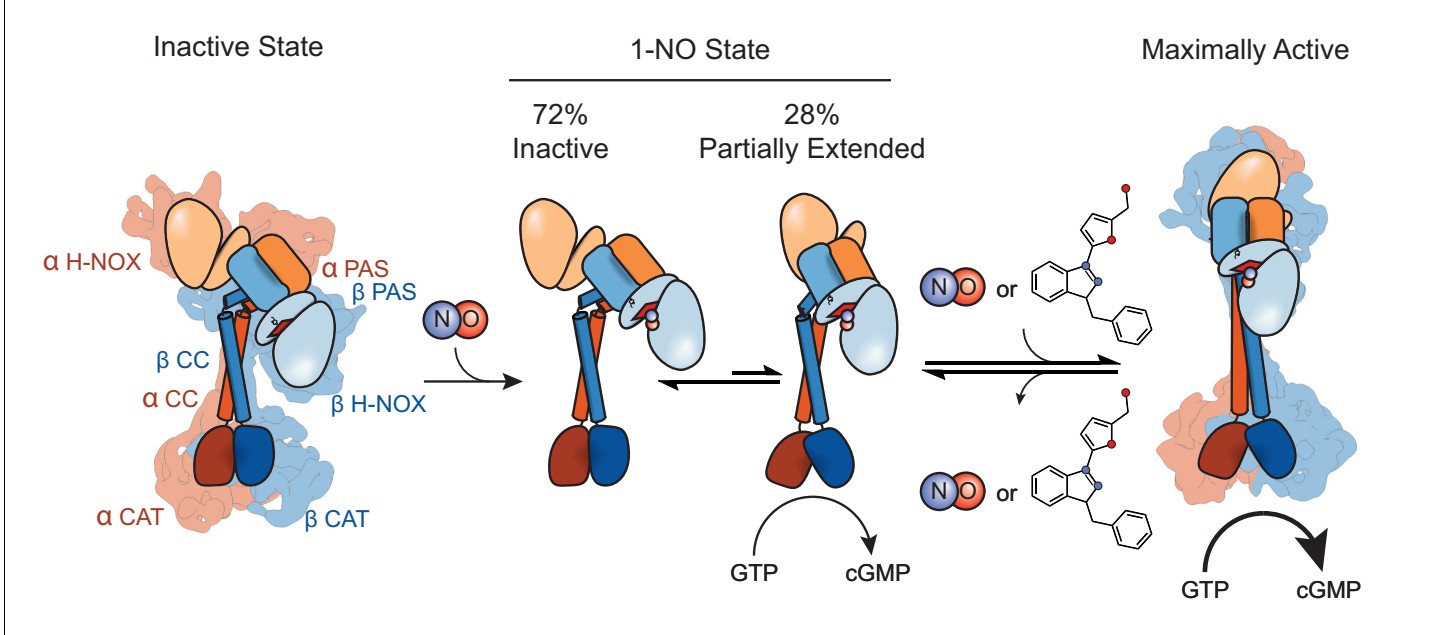

**Figure 6.** Model for conformational rearrangement upon sGC activation. Schematic of the sGC activation pathway. sGC adopts a bent CC conformation after formation of the holoenzyme. Upon the addition of NO an equilibrium exists between the inactive and partially extended states, conferring partial activation of sGC. When excess NO enters the cell or the addition of a stimulator compound, the equilibrium shifts to the active state, allowing for an open CAT dimer conformation and maximal activity.

DOI: https://doi.org/10.7554/eLife.50634.040

## Construction of plasmids

*Manduca sexta* sGC α1 (Uniprot: O77105) and β1(Uniprot: O77106) genes were purchased as gBlocks from IDT with a C-terminal 6x His tag on the α1 gene with Golden Gate cloning sites. The gBlocks were subcloned into a Golden Gate entry vector (gift of the Tullman-Ercek lab, Northwestern University). PCR was used to add regions of homology to pFastBac (Bac-to-Bac Baculovirus Expresion System, Thermo Fisher Scientific), and Gibson Assembly was performed to generate pFastBac_*Ms*_sGC_α1_His6 and pFastBac_*Ms*_sGC_β1. The genes were transposed into a baculovirus bacmid using DH10Bac-GFP cells. The bacmid was isolated, validated, and then transfected into SF9 cells (Berkeley Cell Culture Facility) to generate recombinant baculovirus for both genes.

| Name | Sequence (5' → 3') | Target | Purpose |
|---|---|---|---|
| oBGH130 | GGAGATAATTAAAATGATAACCATCTCGC | pFastBac | Sequencing |
| oBGH131 | TTTATTTGTGAAATTTGTGATGCTATTGC | pFastBac | Sequencing |
| oBGH297 | CGATGGAGTACGAAGCGAATTTCGTATG | Ms_α | Sequencing |
| oBGH168 | GCCTGCACGATCATCTCGGGAC | Ms_β | Sequencing |
| oBGH195 | CTGCCAAGCTCCAGGAACACC ATGACGTGTCCATTCCGTCGTG | Ms_α (F) | Cloning |
| oBGH196 | CCTCGAGACTGCAGGCTCTAGA TTAGTGATGGTGATGGTGATG | Ms_α (R) | Cloning |
| oBGH197 | GCTCCGGGAGCGGACACC ATGTACGGGTTTGTGAACTATGCCC | Ms_β (F) | Cloning |
| oBGH198 | CCCCACAAAGGCTGCCAT TTAATGGATCTTCCTGGTGAGGAACCAG | Ms_β (R) | Cloning |
| oBGH199 | CATCACCATCACCATCACTAATCTAGAGCCTGCAGTCTCGAGG | pFastBac (F) | Cloning |
| oBGH200 | CACGACGGAATGGACACGTCATGGTGTTCCTGGAGCTTGGCAG | pFastBac (R) | Cloning |
| oBGH201 | CTGGTTCCTCACCAGGAAGATCCATTAAATGGCAGCCTTTGTGGGG | pFastBac (F) | Cloning |
| oBGH202 | GGGCATAGTTCACAAACCCGTACATGGTGTCCGCTCCCGGAGC | pFastBac (R) | Cloning |

## Protein expression and purification

SF9 cells were maintained in monolayer and in suspension in ExCell-420 media at 27°C (cells were shaken at 135 rpm). The recombinant baculovirus was amplified until the titer was greater than $1 \times 10^8$ cfu/mL. Five liters of SF9 cells were coinfected with 50 mL of amplified virus per liter and allowed to express for 72 hr. Cells were spun down at 4300 $g$ for 20 min, snap frozen in liquid nitrogen, and stored at –80°C.

The following steps were performed at 4°C. Cell pellets were thawed in ice water and resuspended in lysis buffer: Buffer A (50 mM $Na_2HPO_4$, pH 8.0, 200 mM NaCl, 1 mM imidazole, 1 mM benzamidine, 5% (v/v) glycerol, 0.22 μm filtered) supplemented with 10 mM benzamidine, 1 mM AEBSF, 0.5 mg/mL bovine DNAse I, and 5 mM β-mercaptoethanol. Cells were lysed in a bead beater (BioSpec) with 0.5 mm glass beads. The resulting cell lysate was clarified by spinning at 4300 $g$ for 5 min, followed by spinning at 158,000 $g$ for 2 hr (Ti45 rotor, Beckman Coulter). The lysate was passed through a column containing 2 mL TALON superflow at 0.5 mL/min, and the flow-through was collected. The column was washed with 15 column volumes of Buffer A supplemented with 5 mM β-mercaptoethanol at 0.5 mL/min. The protein was eluted with 10 CV of Buffer B (50 mM $Na_2HPO_4$, pH 8.0, 200 mM NaCl, 150 mM imidazole, 1 mM benzamidine, 5% (v/v) glycerol, 0.22 μm filtered) supplemented with 5 mM β- mercaptoethanol. Fractions with yellow color were concentrated to <2 mL using a 30,000 molecular weight cutoff spin concentrator, supplemented with 5 mM DTT and 5 mM EDTA, and stored overnight. Next, the sample was diluted to 9 mL with Buffer C (25 mM triethanolamine, 25 mM NaCl, 5 mM DTT, 5% (v/v) glycerol, 0.22 μm filtered), and applied to a POROS HQ2 anion exchange column (Thermo Fisher Scientific). The column was washed with 3 CV of Buffer C, and then a gradient to 50% Buffer D (25 mM triethanolamine, 750 mM NaCl, 5 mM DTT, 5% (v/v) glycerol, 0.22 μm filtered) was established over 17 CV at 0.5 mL/min. Fractions with purified sGC were concentrated to 5–50 μM and stored in liquid nitrogen. A typical yield for this expression and purification procedure is 100 μg sGC per liter of insect cells.

## Intact protein mass spectrometry

Purified proteins were buffer-exchanged into 25 mM HEPES, pH 7.5, 25 mM NaCl using three rounds of dilution and concentrations in a Vivaspin 500 (30,000 MWCO) spin concentrator. Samples were filtered with a 0.22 μM spin filter (Millipore). The final sample concentration was approximately 5 μM. Samples were separated with an Agilent 1200 series high-pressure liquid chromatography (HPLC) system over a ProSwift column (ThermoFisher Scientific), and subsequently analyzed by an Agilent 6224 time-of-flight (TOF) mass spectrometer with a Turbospray ion source in positive ion mode.

## Absorption spectroscopy

Samples were reduced in an anaerobic glove bag (Coy) with 5 mM $Na_2S_2O_4$ for 15 min, and then desalted with a BioSpin6 column equilibrated with Buffer E (50 mM HEPES, pH 7.5, 150 mM NaCl, 5% (v/v) glycerol, 0.22 μm filtered). CO-saturated buffer (950 μM CO) was prepared by sparging 3 mL of anaerobic Buffer E for 15 min in a Reacti-Vial (Thermo Fisher Scientific). CO was added to the sample to achieve a final concentration of 425 μM. Nitric oxide was added to a concentration of 500 μM by addition of DEA NONOate, based on 1.5 moles of NO released per mole of NONOate. Protein-ligand complexes were incubated for 15 min at room temperature to establish equilibrium, and no further spectral changes were observed after this time. Samples were placed in a septum-sealed 1 cm pathlength quartz cuvette inside the glove bag, and UV–Vis spectra were recorded on a Cary 300 spectrophotometer (Agilent Technologies).

## Activity assays and quantification

Steady-state kinetics for *Ms* sGC were measured by quantifying the amount of cGMP produced in duplicate endpoint activity assays, performed in at least biological triplicate. Samples were reduced in an anaerobic glove bag with 5 mM $Na_2S_2O_4$ for 15 min, and then desalted with a BioSpin6 column equilibrated with Buffer E. *Ms* sGC with 1-NO and xsNO were prepared by first adding PROLI NONOate to 50 μM, based on 2 moles of NO released per mole of PROLI NONOate. This sample was

then buffer exchanged into Buffer E through a BioSpin6 column to generate the 1-NO state. To generate the xsNO state, PROLI NONOate was added back to a portion of the 1-NO sample and allowed to equilibrate for 5 min. YC-1 was added from a 100x stock solution in DMSO to a final concentration of 150 µM (final DMSO concentration 1%). The protein concentration was determined using the reduced heme Soret peak at 433 nm (149,000 M$^{-1}$ cm$^{-1}$) (*Hu et al., 2008*). Protein concentration was adjusted after desalting the excess NO by comparing the A$_{280}$ peaks to the unliganded protein. Activity assays were conducted at 25°C and pH 7.5 in Buffer D, supplemented with 5 mM DTT and 5 mM MgCl$_2$. Reactions were initiated with 2 mM GTP and timepoints from were quenched with 125 mM Zn(CH$_3$CO$_2$)$_2$, followed by 125 mM Na$_2$(CO$_3$) to adjust the pH to 10.5. Samples were frozen at –80°C until quantification. Quenched assays were thawed, and the zinc precipitate was spun down for 10 min at 23,150 *g*. The reactions were diluted by one to three orders of magnitude, and the cGMP was quantified in duplicate using an extracellular cGMP Enzyme Linked Immunosorbent Assay, following the manufacturer's instructions (Enzo Life Sciences). Concentrations of cGMP were determined from a standard curve, generated over 0.16–500 pmol/mL. Initial rates were calculated from the linear phase of the time course, where 5–10% of the GTP substrate was consumed. The experiment was repeated at least three times to ensure reproducibility.

## Cryo-EM sample preparation and data collection

Samples were reduced in an anaerobic glove bag with 5 mM Na$_2$S$_2$O$_4$ for 15 min, and then desalted with a BioSpin6 column equilibrated with Buffer F (25 mM triethanolamine, pH 7.5, 25 mM NaCl, 5 mM DTT, 0.22 µM filtered). The inactive protein sample was diluted after thawing from a single frozen stock to ~2–4 µM in 1–3% trehalose. The sample (3.5 µL) was applied to glow discharged UltrAUfoil 2/2 200 mesh gold grids (Quantifoil) and plunge-frozen using a vitrobot Mark IV (Thermo Fischer). Blotting was performed under 100% humidity with zero blot force for 3–6 s. sGC was dispersed in vitreous ice which displayed clear contrast transfer function information (*Figure 2—figure supplement 1a*. Active sample was prepared in a similar manner, but 500 µM NO (from DEA NONOate) and 150 µM YC-1 were added (*Figure 3—figure supplement 1a-b*).

Grids were screened and imaged on a Talos Arctica (Thermal Fischer) operated at 200 kV. Complete imaging conditions are described in *Supplementary file 1*. Micrographs were collected at 36,000X nominal magnification on a K3 direct electron detector (Gatan) in super-resolution counted mode at 0.5685 å/pix. Serial EM was used for automated image shift data collection of 2841 and 9330 movies for the inactive and active sample, respectively. Movies were taken in 100 ms frames, totaling an electron dose of 60 electrons per movie.

## Cryo-EM data processing

Inactive sGC movies were drift-corrected, gain-corrected, and binned to 1.137 Å/pix in 7 × 5 patches using MotionCor2 (*Zheng et al., 2017*). Micrographs were CTF-corrected using CTFFIND4 and single particles were manually picked for initial 2D classification within Relion 3.0 (*Rohou and Grigorieff, 2015*; *Scheres, 2012*; *Nakane et al., 2018*). Class averages representing the full-length complex (*Figure 2—figure supplement 1c*) were used for template picking with Relion autopicker, resulting in 675,956 particles. Particles were imported into Cryosparc2 and pruned using 2D classification and 3D *ab initio* classification (*Punjani et al., 2017*). Initial 2D classification yielded a large number of falsely picked background particles and particles which represented broken sGC dimers. We suspect the background particles are due to the size of the complex, while the broken particles likely stem from damage at the air-water interface (*Figure 2—figure supplement 1c*). These poor class averages were removed from further processing steps. In total, 59,165 final particles were refined using non-uniform refinement with default parameters (*Figure 2—figure supplement 1b*). Active sGC movies were binned to 2.274 Å/pix before template-free Laplacian picking for initial particle selection (*Nakane et al., 2018*). Micrographs were manually cleaned by visual inspection and FFT quality. Initial single particles were pruned using iterative 2D and 3D techniques (*Figure 3—figure supplement 1b, c*), as described above for the inactive state. Similarly, many of the initial picked particles were background or broken in the Active dataset (*Figure 3—figure supplement 1c*) Blurred density at low thresholds was seen in a predicted region for the alpha-HNOX but did not resolve during processing and was masked away during the final reconstruction. Of note, both Inactive and Active datasets underwent exhaustive processing schemes, including Bayesian polishing,

multibody refinement and focused classification, none of which improved the resolution or showed evidence of multiple conformations in a single dataset.

## Model building

Homology models for each domain were built using *Phyre2* and correspond to the following PDB entries: α H-NOX (2O0C), α PAS (4GJ4), α CC (3HLS), α CAT (3UVJ), β H-NOX (2O0C), β PAS (4GJ4), β CC (3HLS), β CAT (2WZ1) (*Kelley et al., 2015*). Domains (with side chains removed) were rigid-body docked into the inactive reconstruction using the *fit_in_map* function of Chimera (*Pettersen et al., 2004*). Clear density for heme in the β H-NOX domain enabled clear distinction of the two H-NOX domains. Linkers between the H-NOX and PAS domains are missing in the density, likely due to flexibility, and were left out of the model (α 238–278 and β184–206). Initial placement of the CAT dimer was based on a significant extension in the sequence of the α CAT C-terminus, which is visible as unmodeled density. Continuous density from the CAT dimer enabled assignment of the CC and PAS domains. The linker region between the PAS domains and the CC were manually modeled in COOT based on secondary structure predictions from PSIPRED (*Emsley et al., 2010*; *Vynne, 1997*). Refinement using iterative rounds of *phenix.real_space_refine* and inspection in COOT led to the final carbon back bone trace of the inactive state. Active state modeling was based on rigid-body fitting of domains based on the inactive state. Helical density was apparent through-out the structure (with the exception of the α H-NOX, as mentioned above). Linker regions between domains were corrected with COOT and *phenix.real_space_refine*. Final model idealization was carried out using *phenix.model_*idealization and validated using Molprobity (*Chen et al., 2010*).

## Small-angle X-ray scattering in-line with size-exclusion chromatography (SEC-SAXS)

Samples were reduced in an anaerobic glove bag with 5 mM $Na_2S_2O_4$ for 15 min, concentrated to 60 μL at 50 μM (~7.5 mg/mL), and then three rounds of dialysis were performed in Buffer G (50 mM $KH_2PO_4$, pH 7.4, 150 mM NaCl, 2% glycerol). The activated sample was prepared with 500 μM NO (from DEA NONOate). The samples were sealed and run within 3 hr of being prepared; samples did not undergo freeze–thaw after the sample was prepared (the protein was freeze–thawed once after purification for initial storage).

In situ sample purification was accomplished through SEC to isolate well-folded proteins from aggregates and other impurities immediately before exposure to synchrotron X-ray radiation using a custom designed flow cell. SEC-SAXS was collected at the SIBYLS beamline (bl12.3.1) at the Advanced Light Source at Lawrence Berkeley National Laboratory, Berkeley, California (*Classen et al., 2010*; *Classen et al., 2013*). X-ray wavelength was set at λ = 1.127 Å and the sample-to-detector distance was 2105 mm, as determined by silver behenate calibration, resulting in scattering vectors, q, ranging from 0.01 $Å^{-1}$ to 0.4 $Å^{-1}$. The scattering vector is defined as q = 4πsinθ/λ, where 2θ is the scattering angle. Data was collected using a Dectris PILATUS3 × 2M detector at 20°C and processed as previously described (*Dyer et al., 2014*; *Hura et al., 2009*). Briefly, a custom-made SAXS flow cell was directly coupled with an Agilent 1260 Infinity HPLC system using a Shodex KW-803 column. The column was equilibrated with running Buffer E with a flow rate of 0.5 mL/min for inactive sGC and 0.55 mL/min for activated sGC. To achieve activation conditions, the buffer was continuously sparged with nitrogen gas and the column was equilibrated for at least 2 hr to maintain an anaerobic environment. Several NONOates with various half-lives were added to the running buffer to achieve activation of sGC. These NONOates include DEA NONOate and DETA NONOate, which spontaneously release NO with half-lives of 16 min and 56 hr, respectively. Each sample was run through the SEC-SAXS system and 3 s X-ray exposures were collected continuously over the 30 min elution. The SAXS frames recorded prior to the protein elution peak were used to correct all other frames. The corrected frames were investigated by radius of gyration $R_g$ derived by the Guinier approximation I(q)=I(0) exp(–$q^2R_g^2$/3) with the limits q*$R_g$ <1.3 (*Figure 5—figure supplement 2*). The elution peak was mapped by comparing the integral of ratios to background and $R_g$ relative to the recorded frame using the program SCÅTTER (*Figure 5—figure supplement 1*).

The frames in the regions of least $R_g$ variation were averaged and merged in SCÅTTER to produce the highest signal-to-noise SAXS curves. These merged SAXS curves were used to generate

the Guinier plots, volumes-of-correlation (V$_c$), pair distribution functions, P(r), and normalized Kratky plots. The Guinier plot indicated an aggregation-free state of the protein (*Figure 5—figure supplement 2*). The P(r) function was used to determine the maximal dimension of the macromolecule (D$_{max}$) and estimate inter-domain distances (*Figure 4A*) (*Putnam et al., 2007*). P(r) functions were normalized based on the molecular weight (MW) of the assemblies, as determined by the calculated V$_c$ (*Rambo and Tainer, 2013*).

Eluent was subsequently split (4 to 1) between the SAXS line and a multiple wavelength detector (UV-vis), set to 432 and 280 nm, multi-angle light scattering (MALS), and refractometer. The ratios of the protein (280 nm) and Soret band (432 nm) of the heme from SEC were used to evaluate the ligation state of *Ms* sGC upon NO binding (*Figure 5—figure supplement 1*). MALS experiments were performed using an 18-angle DAWN HELEOS II light scattering detector connected in tandem to an Optilab refractive index concentration detector (Wyatt Technology). System normalization and calibration was performed with bovine serum albumin using a 60 µL sample at 10 mg/mL in the same SEC running buffer and a dn/dc value of 0.185 and 0.15 mL/g for inactive and active sGC respectively. The MALS data was used to compliment the MWs calculated by the SAXS analysis and, being furthest downstream, the MALS peaks were used to align the SAXS and UV-vis peaks along the x-axis (elution volume in mL/min) to compensate for fluctuations in timing and band broadening (*Figure 5—figure supplement 1*). UV–vis data was integrated using Agilent Chemstation software and baseline corrected using Origin 9.1 (*Figure 5—figure supplement 1*). MALS and differential refractive index data was analyzed using Wyatt Astra seven software to monitor the homogeneity of the sample molecular weights across the elution peak, complementing the SEC-SAXS signal validation (*Figure 5—figure supplement 1*).

## Solution structure modeling

The cryo-EM refined crystal structure for inactive sGC was used to build an initial atomistic model; all missing loops and terminal residues were added using MODELLER (*Fiser et al., 2000*). A purpose-designed rigid body modeling pipeline was applied to the completed structure using BIL-BOMD to systematically explore conformational space for both the inactive and activated states of sGC (*Video 4*) (*Pelikan et al., 2009*). To obtain an inactive state model, the α H-NOX domain and both α terminal tails were moved, then all α and β domains were moved by allowing flexibility at the hinge region where the CCs meet the CAT domains, while holding the CAT domains fixed and still permitting flexibility of the terminal tails. This was then used to generate the 1-NO state model by moving the same regions as the inactive state in reverse order. Theoretical SAXS curves were produced by FOXS (*Schneidman-Duhovny et al., 2010*; *Schneidman-Duhovny et al., 2013*). Each model generated through BILBOMD as compared to the experimental SAXS profiles to assess the goodness of fit. Multistate model ensembles for activated sGC were determined using MultiFOXS (*Schneidman-Duhovny et al., 2016*).

## Acknowledgements

The authors thank members of the Marletta Lab and SN Gates for discussions and critical reading of this manuscript. Access to the FEI Talos Arctica was provided through BACEM UCB Facility. This work was made possible by support through the National Institutes of Health (NIH) grant (1R01GM127854) to MAM. BGH was supported in part by a NIH Chemistry-Biology Interface Institutional Training Grant (T32 GM066698). ALY is a fellow of the Jane Coffin Childs Memorial Fund for Medical Research. SAXS data was collected at the Advanced Light Source (ALS), SIBYLS beamline on behalf of US DOE-BER, through the Integrated Diffraction Analysis Technologies (IDAT) program. Additional support comes from the NIGMS project ALS-ENABLE (P30 GM124169) and a High-End Instrumentation Grant S10OD018483.

## Additional information

### Competing interests

Michael A Marletta: Reviewing editor, *eLife*. The other authors declare that no competing interests exist.

## Funding

| Funder | Grant reference number | Author |
| --- | --- | --- |
| National Institutes of Health | GM127854 | Michael A Marletta |
| National Institutes of Health | GM066698 | Benjamin G Horst |
| National Institutes of Health | GM124169 | Michal Hammel Daniel J Rosenberg |
| Jane Coffin Childs Memorial Fund for Medical Research | | Adam L Yokom |
| National Institutes of Health | S10OD018483 | Michal Hammel Daniel J Rosenberg |

The funders had no role in study design, data collection and interpretation, or the decision to submit the work for publication.

## Author contributions

Benjamin G Horst, Conceptualization, Validation, Investigation, Visualization, Methodology, Writing—original draft, Writing—review and editing; Adam L Yokom, Formal analysis, Validation, Investigation, Visualization, Methodology, Writing—original draft, Writing—review and editing; Daniel J Rosenberg, Formal analysis, Investigation, Visualization, Methodology, Writing—review and editing; Kyle L Morris, Conceptualization, Investigation, Methodology, Writing—review and editing; Michal Hammel, Formal analysis, Visualization, Methodology, Writing—review and editing; James H Hurley, Conceptualization, Resources, Formal analysis, Validation, Visualization, Methodology, Project administration, Writing—review and editing; Michael A Marletta, Conceptualization, Supervision, Funding acquisition, Validation, Methodology, Project administration, Writing—review and editing

## Author ORCIDs

Benjamin G Horst (iD) https://orcid.org/0000-0001-9694-7263
Adam L Yokom (iD) https://orcid.org/0000-0002-3746-7961
Daniel J Rosenberg (iD) https://orcid.org/0000-0001-8017-8156
Kyle L Morris (iD) https://orcid.org/0000-0002-1717-8134
James H Hurley (iD) https://orcid.org/0000-0001-5054-5445
Michael A Marletta (iD) https://orcid.org/0000-0001-8715-4253

## Decision letter and Author response

Decision letter https://doi.org/10.7554/eLife.50634.054
Author response https://doi.org/10.7554/eLife.50634.055

# Additional files

## Supplementary files

• Supplementary file 1. Cryo-EM data acquisition, image processing and model refinement.
DOI: https://doi.org/10.7554/eLife.50634.043

• Supplementary file 2. SEC-SAXS-MALS-UV-visible absorption results for the activation of sGC.
DOI: https://doi.org/10.7554/eLife.50634.044

• Transparent reporting form DOI: https://doi.org/10.7554/eLife.50634.045

## Data availability

Maps have been deposited to EMDB (20282, 20283) as well the C-alpha traces are deposited to the PDB (6PAS, 6PAT) for the inactive and active states. SAXS curves deposited to SASBDB.

The following datasets were generated:

| Author(s) | Year | Dataset title | Dataset URL | Database and Identifier |
|---|---|---|---|---|
| Horst BG, Yokom AL, Rosenberg DJ, Morris KL, Hammel M, Hurley JH, Marletta MA | 2019 | Single particle reconstruction 5.1Å resolution | http://www.ebi.ac.uk/pdbe/entry/emdb/EMD-20282 | Electron Microscopy Data Bank, EMD-20282 |
| Horst BG, Yokom AL, Rosenberg DJ, Morris KL, Hammel M, Hurley JH, Marletta MA | 2019 | Single particle reconstruction 5.8Å resolution | http://www.ebi.ac.uk/pdbe/entry/emdb/EMD-20283 | Electron Microscopy Data Bank, EMD-20283 |
| Horst BG, Yokom AL, Rosenberg DJ, Morris KL, Hammel M, Hurley JH, Marletta MA | 2019 | C-alpha traces | http://www.rcsb.org/structure/6PAS | Protein Data Bank, 6PAS |
| Horst BG, Yokom AL, Rosenberg DJ, Morris KL, Hammel M, Hurley JH, Marletta MA | 2019 | C-alpha traces | http://www.rcsb.org/structure/6PAT | Protein Data Bank, 6PAT |

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
