## [Decision Letter]

Thank you for submitting your article "Allosteric activation of the nitric oxide receptor soluble guanylate cyclase mapped by cryo-electron microscopy" for consideration by *eLife*. Your article has been reviewed by three peer reviewers, and the evaluation has been overseen by Wilfred van der Donk as Reviewing Editor and Philip Cole as the Senior Editor. The following individuals involved in review of your submission have agreed to reveal their identity: Andrea Mattevi (Reviewer #1); Annie Beuve (Reviewer #2).

The reviewers have discussed the reviews with one another and the Reviewing Editor has drafted this decision to help you prepare a revised submission.

Summary:

This study presents two cryo-EM structures of the soluble guanylyl cyclase (sGC), the primary receptor of nitric oxide in mammals. The authors report the inactive structure in the absence of nitric oxide, and an active structure in the presence of nitric oxide and a stimulating molecule YC-1. The data address a long-standing question in the field of NO-sGC signaling related to how the NO signal propagates from the heme receptor domain to the effector catalytic domain. The findings in this work are of high (patho)physiological importance as sGC is a key target for cardiovascular diseases.

Determination of the molecular mechanism of activation has been impaired by the lack of a full-length structure of this heterodimeric enzyme formed by two subunits, each containing 4 domains (HNOX, PAS, coiled-coil, catalytic). The current work provides the basal enzyme structure at 5.8 Å and the activated enzyme at 5.1 Å resolution. The analysis relies on a single 3D class arising from less than 6% of the total particles that were identified on the grid, corresponding to 25000-40000 particles effectively used for the density reconstruction. As a consequence, the maps are only at medium resolution, but the conformational changes are large and hence readily and convincingly identified. The authors are careful not to overinterpret their data as the authors limit their interpretation to the existence of multiple conformations in the partly activated enzyme.

Upon activation, a helical connecting domain straightens to trigger large rotations of the heme-binding domain and PAS domains that lead to the opening of the active site in the catalytic domain. The enzyme thereby acquires the ability to bind GTP. This suggests a new paradigm of activation not only for sGC but also for S-helix containing proteins. A potential site for the stimulator YC-1 is also identified, and the findings rebut some long-established beliefs in the molecular mechanism of activation of sGC. Thus, this work will provide invaluable resources for many groups and indicate routes to pursue in future research.

Although the reviewers are generally enthusiastic about the findings and their publication in *eLife*, some concerns and questions were raised that the authors should address prior to publication:

Essential revisions:

1) The authors should provide more details about the cryo-EM data processing procedures. Were other classes identified? Were the rejected particles (>90% of the total) really inconclusive and uninformative? What was the issue? Disorder? Multiple conformations? Are they due to too inclusive particle picking, damaged particles at the air water interface or conformational variability? Considering that this manuscript describes an allosteric mechanism, it is critical that the cryo-EM data processing workflow is better described, and if necessary re-processed, to quantitatively assess the heterogeneity in each dataset to support the proposed model where inactive state protein is predominantly in one conformation, YC-1- and NO-bound protein is in predominantly in a second active state conformation, and stochiometric NO-bound protein is fluctuating between those two states.

2) The activated state has been obtained with a combination of nitric oxide and a stimulating agent YC-1. It is therefore not possible to delineate the role of each in the conformational change observed. The small angle X-ray scattering data presented addresses this issue by comparing scatterings of the enzyme in an inactive state and a state activated by nitric oxide only and seem to support a model where the enzyme fluctuates between inactive and active conformations in the presence of nitric oxide and where YC-1 stabilizes the active conformation further, but the experiment is missing an important control, which is scattering data from the same conditions as the active cryo-EM structure. While obtaining cryo-EM data from a sample activated by NO only and SAXS data for a sample activated by NO and YC-1 would greatly strengthen the manuscript and appear very feasible given the data presented, it is understood that, if sample or instrument availability are limiting, collecting such additional data may be outside the scope of a revision. If the collection of such data was attempted but unsuccessful, it should be noted in the manuscript.

3) Along similar lines, conformational variability should be assessed and described in detail. Can enzyme fluctuating between inactive and active states still be observed by SAXS in the presence of YC-1?

4) The figures are generally rather small and sometimes not very clear. For instance, in the last paragraph ofsubsection “Inactive conformation of sGC exhibits bent coiled-coils”, Figure 2E is very unclear and needs to show density to support the statement. In the third paragraph of the subsection “Activated sGC extends the regulatory lobe from catalytic core”, comparison is made between two different figures. It would help to show this in a single figure.

5) The purification of *Ms* sGC-Histag is done by infection of Sf9 insect cells and one of the steps involves elution through a talon column. Although all the data presented in the manuscript (NO-stimulated activity, UV absorbance, heme density in the cryo-EM) suggest heme is present, there is concern that a significant percentage of the purified sGC could be in an apo-form, which would affect some experimental outcomes. Sf9 cells are usually very poor at synthesizing heme and elution of a talon column with 150 mM imidazole could affect the heme (His-bound) of sGC. Maybe the culture or the purified samples were complemented by excess hemin or 5-ALA but this is not mentioned in the Materials and methods. The authors should clarify heme content and the methods used (if any) to ensure maximal heme content. In addition, the heme density is not properly shown in the figure, which should be corrected.

6) The authors need to add text to describe more extensively the YC-1 activator. How was it identified? Affinity? Biological activity? Since an YC-1 analog (riociguat) is a new treatment for pulmonary arterial hypertension, this will be valuable for readers. In addition, a very recent structure of inactivated and NO-bound activated sGC was just published online. The authors of the current work should emphasize the YC-1 angle, with a particular emphasis on its role in the observed allosteric activation, since the online report does not contain such a stimulating agent. The authors of the current work are highly qualified to draw conclusions and interpretations from comparing their structures with YC-1 (even at a lesser resolution) to the newly published structures because of their expertise in sGC mechanisms of activation.

7) The authors have been careful to not over-interpret their data given the limited resolution of their study and the absence of critical controls. Still, these caveats should be discussed thoroughly in the manuscript.

---

## [Author Response]

Essential revisions:1) The authors should provide more details about the cryo-EM data processing procedures. Were other classes identified? Were the rejected particles (>90% of the total) really inconclusive and uninformative? What was the issue? Disorder? Multiple conformations? Are they due to too inclusive particle picking, damaged particles at the air water interface or conformational variability? Considering that this manuscript describes an allosteric mechanism, it is critical that the cryo-EM data processing workflow is better described, and if necessary re-processed, to quantitatively assess the heterogeneity in each dataset to support the proposed model where inactive state protein is predominantly in one conformation, YC-1- and NO-bound protein is in predominantly in a second active state conformation, and stochiometric NO-bound protein is fluctuating between those two states.

This is a critical point of clarification for the conformational changes seen by cryo-EM. Many of the initial particles picked in both data sets were ‘false’ particles of background or broken sGC complexes. The first paragraph of the subsection “Inactive conformation of sGC exhibits bent coiled-coils” and the first paragraph of the subsection “Activated sGC extends the regulatory lobe from catalytic core” are clarified stating the 2D averages were cleaned of poor particles. Additionally, a detailed description is included in the Materials and methods (subsection “Cryo-EM Data Processing”). To further clarify our workflow, the supplementary figures for the processing schemes have been expanded to show an initial round of 2D class averages for both Inactive (Figure 2—figure supplement 3) and Active (Figure 3—figure supplement 3) which show partial complexes.

We tested many processing schemes to determine that only a single conformation was present in each dataset (data not shown). To highlight this, in the Materials and methods we have added “Of note, both Inactive and Active datasets underwent exhaustive processing schemes, including Bayesian polishing, multibody refinement and focused classification, none of which improved the resolution or showed evidence of multiple conformations in a single dataset.”

2) The activated state has been obtained with a combination of nitric oxide and a stimulating agent YC-1. It is therefore not possible to delineate the role of each in the conformational change observed. The small angle X-ray scattering data presented addresses this issue by comparing scatterings of the enzyme in an inactive state and a state activated by nitric oxide only and seem to support a model where the enzyme fluctuates between inactive and active conformations in the presence of nitric oxide and where YC-1 stabilizes the active conformation further, but the experiment is missing an important control, which is scattering data from the same conditions as the active cryo-EM structure. While obtaining cryo-EM data from a sample activated by NO only and SAXS data for a sample activated by NO and YC-1 would greatly strengthen the manuscript and appear very feasible given the data presented, it is understood that, if sample or instrument availability are limiting, collecting such additional data may be outside the scope of a revision. If the collection of such data was attempted but unsuccessful, it should be noted in the manuscript.

We agree with the reviewers that SAXS data with NO and YC-1 would be an excellent addition to the manuscript. Unfortunately, preliminary SAXS data collected with *Ms* sGC + YC-1 and *Ms* sGC + NO + YC-1 indicated that YC-1 is not soluble enough to collect scattering data, and thus a full experiment was not attempted as SAXS is costly in terms of protein required. Even compounds that are further along in their development to riociguat (e.g. BAY 41-2272) are not soluble enough for these experiments. The introductory sentence to the subsection “SAXS shows a distribution of inactive and active states in solution” has been modified to explain why this control is not present.

3) Along similar lines, conformational variability should be assessed and described in detail. Can enzyme fluctuating between inactive and active states still be observed by SAXS in the presence of YC-1?

Please see the response to the previous comment.

4) The figures are generally rather small and sometimes not very clear. For instance, in the last paragraph of subsection “Inactive conformation of sGC exhibits bent coiled-coils”, Figure 2E is very unclear and needs to show density to support the statement. In the third paragraph of the subsection “Activated sGC extends the regulatory lobe from catalytic core”, comparison is made between two different figures. It would help to show this in a single figure.

The comparison between Figure 2D and Figure 3D has been removed for clarity. We feature the comparison between the two structures in Figure 4. Specifically, Figure 4D shows the coiled-coil domains in the inactive and active state.

5) The purification of Ms sGC-Histag is done by infection of Sf9 insect cells and one of the steps involves elution through a talon column. Although all the data presented in the manuscript (NO-stimulated activity, UV absorbance, heme density in the cryo-EM) suggest heme is present, there is concern that a significant percentage of the purified sGC could be in an apo-form, which would affect some experimental outcomes. Sf9 cells are usually very poor at synthesizing heme and elution of a talon column with 150 mM imidazole could affect the heme (His-bound) of sGC. Maybe the culture or the purified samples were complemented by excess hemin or 5-ALA but this is not mentioned in the Materials and methods. The authors should clarify heme content and the methods used (if any) to ensure maximal heme content. In addition, the heme density is not properly shown in the figure, which should be corrected.

While the presence of apo sGC in our preparations is a constant concern for us, the final purified fraction of *Ms* sGC is primarily if not exclusively in the holo from. We know this for several reasons: firstly, in the anion exchange step of our purification (after the IMAC TALON column), apo sGC elutes after the holo enzyme and those fractions are not carried forward in the prep. Also, based on extinction coefficients for full-length sGC purified from bovine lung (Stone and Marletta, 1994) and studies with stand-alone HNOX proteins from bacteria, the ratio of the absorption of the heme Soret band to the absorption of the protein is as expected (for *Ms* sGC, a ratio of 0.8 is expected for 434 nm / 280 nm). For these reasons, as well as the data the reviewer mentions, we are confident in the quality of the preparations of *Ms* sGC.

6) The authors need to add text to describe more extensively the YC-1 activator. How was it identified? Affinity? Biological activity? Since an YC-1 analog (riociguat) is a new treatment for pulmonary arterial hypertension, this will be valuable for readers. In addition, a very recent structure of inactivated and NO-bound activated sGC was just published online. The authors of the current work should emphasize the YC-1 angle, with a particular emphasis on its role in the observed allosteric activation, since the online report does not contain such a stimulating agent. The authors of the current work are highly qualified to draw conclusions and interpretations from comparing their structures with YC-1 (even at a lesser resolution) to the newly published structures because of their expertise in sGC mechanisms of activation.

Two sentences more clearly introducing YC-1 have been included in the third paragraph of the Introduction. We have added three sentences to the third paragraph of the Discussion comparing the xsNO + YC-1 map to the newly published xsNO only map, emphasizing the similarity in map in all regards except for the density we assign to YC-1. While we agree that the potential location of the binding pocket of sGC stimulators is an exciting finding, we feel it would be misleading to highlight the YC-1 in our paper more than this addition and our analysis, given the resolution of our map.

7) The authors have been careful to not over-interpret their data given the limited resolution of their study and the absence of critical controls. Still, these caveats should be discussed thoroughly in the manuscript.

We thank the reviewer for their acknowledgement that we have been careful not to over-interpret our data. We hope that our additions discussing our attempts to improve the resolution, the limited solubility of sGC stimulators, and the comparison with the new higher resolution structure will satisfy the reviewers concerns for the discussion of these caveats.